# Behavioural Interventions in People with Oropharyngeal Dysphagia: A Systematic Review and Meta-Analysis of Randomised Clinical Trials

**DOI:** 10.3390/jcm11030685

**Published:** 2022-01-28

**Authors:** Renée Speyer, Reinie Cordier, Anna-Liisa Sutt, Lianne Remijn, Bas Joris Heijnen, Mathieu Balaguer, Timothy Pommée, Michelle McInerney, Liza Bergström

**Affiliations:** 1Department Special Needs Education, Faculty of Educational Sciences, University of Oslo, 0318 Oslo, Norway; 2Curtin School of Allied Health, Faculty of Health Sciences, Curtin University, Perth, WA 6102, Australia; reinie.cordier@northumbria.ac.uk; 3Department of Otorhinolaryngology and Head and Neck Surgery, Leiden University Medical Centre, 2333 ZA Leiden, The Netherlands; b.j.Heijnen@lumc.nl; 4Department of Social Work, Education and Community Wellbeing, Faculty of Health & Life Sciences, Northumbria University, Newcastle upon Tyne NE7 7XA, UK; 5Critical Care Research Group, The Prince Charles Hospital, Brisbane, QLD 4032, Australia; annaliisasp@gmail.com; 6School of Medicine, University of Queensland, Brisbane, QLD 4072, Australia; 7School of Allied Health, HAN University of Applied Sciences, 6525 EN Nijmegen, The Netherlands; lianne.remijn@han.nl; 8IRIT, CNRS, Université Paul Sabatier, 31400 Toulouse, France; mathieu.balaguer@irit.fr (M.B.); timothy.pommee@irit.fr (T.P.); 9School of Allied Health (SoAH), Australian Catholic University (ACU), Sydney, NSW 2060, Australia; michelle.mcInerney@acu.edu.au; 10Remeo Stockholm, 128 64 Stockholm, Sweden; liza.bergstrom@regionstockholm.se; 11Speech Therapy Clinic, Danderyd University Hospital, 182 88 Stockholm, Sweden

**Keywords:** deglutition, swallowing disorders, RCT, intervention, compensation, rehabilitation

## Abstract

Objective: To determine the effects of behavioural interventions in people with oropharyngeal dysphagia. Methods: Systematic literature searches were conducted to retrieve randomized controlled trials in four different databases (CINAHL, Embase, PsycINFO, and PubMed). The methodological quality of eligible articles was assessed using the Revised Cochrane risk-of-bias tool for randomised trials (RoB 2), after which meta-analyses were performed using a random-effects model. Results: A total of 37 studies were included. Overall, a significant, large pre-post interventions effect size was found. To compare different types of interventions, all behavioural interventions and conventional dysphagia treatment comparison groups were categorised into compensatory, rehabilitative, and combined compensatory and rehabilitative interventions. Overall, significant treatment effects were identified favouring behavioural interventions. In particular, large effect sizes were found when comparing rehabilitative interventions with no dysphagia treatment, and combined interventions with compensatory conventional dysphagia treatment. When comparing selected interventions versus conventional dysphagia treatment, significant, large effect sizes were found in favour of Shaker exercise, chin tuck against resistance exercise, and expiratory muscle strength training. Conclusions: Behavioural interventions show promising effects in people with oropharyngeal dysphagia. However, due to high heterogeneity between studies, generalisations of meta-analyses need to be interpreted with care.

## 1. Introduction

Swallowing disorders, or oropharyngeal dysphagia (OD), can be the result of many underlying conditions such as stroke, progressive neurological diseases, and acquired brain injury. They may also be the consequence of treatment side effects; for example, radiation or surgical interventions in patients with head and neck oncological disorders. Prevalence of OD in the general population ranges from 2.3 to 16% [1]. However, depending on underlying disease severity and outcome measures used (e.g., instrumental assessment, screening or patient self-report) [2], prevalence estimates can be as high as 80% in stroke and Parkinson’s disease patients, up to 30% in traumatic brain injury patients, and over 90% in patients with community-acquired pneumonia [3]. Also, pooled prevalence estimates for swallowing problems in people with cerebral palsy determined by meta-analyses are as high as 50.4% [4].

OD may have severe effects on a person’s health as dysphagia can lead to dehydration, malnutrition, and aspiration pneumonia. OD also has a high disease burden and poses a major societal challenge, which is associated with significant psychological and social burden, resulting in reduced quality-of-life for both patients and caregivers [5].

The treatment of OD may include surgical, pharmacological and behavioural interventions. Behavioural interventions include: bolus modification and management (e.g., adjusting the viscosity, volume, temperature and/or acidity of food and drinks), motor behavioural techniques or oromotor exercises, general body and head postural adjustments, swallowing manoeuvres (e.g., manoeuvres to improve food propulsion into the pharynx and airway protection), and sensory and neurophysiologic stimulation (e.g., neuromuscular electrical stimulation [NMES]) [6].

An increasing number of reviews have been published over the last two decades on the treatment effects of behavioural interventions in people with OD. However, only one systematic review [7] summarised the effects of swallowing therapy as applied by speech and language therapists without restrictions on subject populations or study designs. Furthermore, while most reviews have focussed on selected types of interventions and patient populations, very few reviews use criteria related to study designs (e.g., [8,9] solely including randomised controlled trials [RCTs], ranked as the highest level of evidence [10]).

This systematic review aimed to determine the effects of behavioural interventions in people with OD based on the highest level of evidence (RCTs) only. Behavioural interventions comprised any intervention by a dysphagia expert, excluding surgical and pharmacological interventions. Clinicians being referred to as dysphagia experts include speech therapists, occupational therapists, or physiotherapists, but may incorporate other disciplines depending on national healthcare and education systems. Finally, neurostimulation techniques were considered out of scope of this current review.

## 2. Methods

The methodology and reporting of this systematic review were based on the Preferred Reporting Items for Systematic Reviews and Meta-Analyses (PRISMA) statement and checklist. The PRISMA 2020 statement and checklist (Appendix A) aim to enhance the essential and transparent reporting of systematic reviews [11,12]. The protocol for this review was registered at PROSPERO, the international prospective register of systematic reviews (registration number: CRD42020179842).

### 2.1. Information Sources

To identify studies, literature searches were conducted on 6 March 2021, across these four databases: CINAHL, Embase, PsycINFO, and PubMed. Publications dates ranged from 1937–2021, 1902–2021, 1887–2021, and late 1700s–2021, respectively. Additional searches included checking the reference lists of eligible articles.

### 2.2. Search Strategies

Electronic search strategies were performed in all four databases using subheadings (e.g., MeSH and Thesaurus terms) and free text terms. Two strings of terms were combined: (1) dysphagia and (2) randomised controlled trial. The full electronic search strategies are reported in Table 1.

### 2.3. Inclusion and Exclusion Criteria

The following criteria for inclusion were applied: (1) participants had a diagnosis of OD; (2) behavioural interventions were aimed at reducing swallowing or feeding problems; (3) studies included a comparison group; (4) participants were randomly assigned to one of the study arms or groups; (5) studies were published in English.

Studies focussing on drooling, self-feeding, gastro-oesophageal reflux or oesophageal dysphagia (e.g., dysphagia resulting from oesophageal carcinoma or esophagitis) were excluded. Further excluded studies were those describing drug-induced swallowing problems, temporary swallowing problems caused by oedema post-surgery (e.g., anterior cervical discectomy), or swallowing problems associated with adverse effects of interventions such as inflammation and oedema resulting from recent radiotherapy (≤three months after intervention) or thyroidectomy. Studies reporting solely on feeding tube removal after intervention that did not provide data on swallowing or feeding problems, were also excluded. Studies on behavioural eating problems including bulimia, anorexia, and picky eaters, were out of scope of this review. Finally, only original research was included, thus excluding, for example, conference abstracts, doctoral theses and reviews.

### 2.4. Systematic Review

*Methodological Quality and Risk of Bias.* The Revised Cochrane risk-of-bias tool for randomised trials (RoB 2) [13] was used to assess the methodological quality of the included studies. The RoB 2 tool provides a framework for evaluating the risk of bias in the findings of any type of randomised trial. The tool is structured along five domains through which bias might be introduced into the study results: (1) the randomisation process; (2) deviations from intended interventions; (3) missing outcome data; (4) measurement of the outcome; (5) selection of the reported result.

*Data Collection Process.* A data extraction form was created to extract data from the included studies under the following categories: methodological quality, participant diagnosis, inclusion criteria, sample size, age, gender, intervention goal, intervention agent/delivery/dosage, intervention condition, outcome measures and treatment outcome.

*Data, Items and Synthesis of Results.* Two independent raters reviewed all titles and abstracts, then original articles, for eligibility. Inclusion of studies was based on consensus between raters. To ensure rating accuracy, two group sessions were held to discuss ratings of one hundred randomly selected records to achieve consensus before rating the remaining abstracts. Where consensus could not be reached between the first two raters, a third party was consulted for resolution. Methodological quality assessment was also rated by two independent researchers, after which consensus was reached with involvement of a third reviewer, when necessary. No evident bias in article selection or methodological study quality rating was present as none of the reviewers had formal or informal affiliations with any of the authors of the included studies. At this stage reviewers did not exclude studies based on type of intervention (e.g., behavioural intervention, neurostimulation).

During data collection, data points across all studies were extracted using comprehensive data extraction forms. Risk of bias was assessed per individual study using RoB 2 [13]. The main summary measures for assessing treatment outcome were effect sizes and significance of findings.

### 2.5. Meta-Analysis

Data was extracted from relevant studies to compare the effect sizes for the following: (1) pre-post outcome measures of OD and (2) mean difference in outcome measures from pre to post between different types of behavioural interventions. All interventions were categorised into compensatory (e.g., body and postural adjustments, or bolus modification), rehabilitative (e.g., oromotor exercises or Shaker exercise), combined compensatory and rehabilitative interventions, and no dysphagia intervention. Only studies using instrumental assessment (videofluoroscopic swallow study [VFSS] or fiberoptic endoscopic evaluation of swallowing [FEES]) to confirm OD were included. Outcome measures based on visuoperceptual evaluation of instrumental assessment and clinical non-instrumental assessments, were eligible for inclusion in meta-analyses. However, if both types of data were available, instrumental assessment was preferred over non-instrumental assessment outcome data. Oral intake measures, screening tools and patient self-report measures were excluded from meta-analyses. Measures other than the authors’ primary outcomes may have been selected if these measures helped to reduce heterogeneity between studies.

To compare effect sizes, group means, standard deviations, and sample sizes for pre- and post-measurements were entered into Comprehensive Meta-Analysis Version 3.3.070 [14]. If only non-parametric data were available (i.e., medians, interquartile ranges), then data were converted into parametric data for meta-analyses. Participants in studies of multiple intervention groups were analysed separately. Where studies used the same participants, only one study was included in the meta-analysis. If studies provided insufficient data for meta-analyses, authors were contacted by e-mail for additional data.

Effect sizes were calculated in Comprehensive Meta-Analysis using a random-effects model. Due to variations in participant characteristics, intervention approaches, and outcome measurements, studies were unlikely to have similar true effects. Heterogeneity was estimated using the *Q* statistic to determine the spread of effect sizes about the mean and *I*^2^ was used to estimate the ratio of true variance to total variance. *I*^2^-values of less than 50%, 50% to 74%, and higher than 75% indicate low, moderate, and high heterogeneity, respectively [15]. Using the Hedges *g* formula for standardized mean difference with a confidence interval of 95%, effect sizes were calculated and interpreted using Cohen’s *d* convention: *g* ≤ 0.2 as no or negligible effect; 0.2 < *g* ≤ 0.5 as minor effect; 0.5 < *g* ≤ 0.8 as moderate effect; and *g* > 0.8 as large effect [16].

Forest plots of effect sizes for OD outcome scores were generated for pre-post behavioural interventions. Due to blended configurations of intervention groupings across studies it was not possible to compare a homogenous behavioural intervention group against a comparison group that did not have a behavioural component. For this reason, only a subgroup between group analysis was conducted (and not an overall between group analysis) to explore effect sizes as a function of various moderators. Behavioural interventions (compensatory, rehabilitative, or combined compensatory and rehabilitative interventions) were compared with conventional dysphagia treatment (CDT), or no dysphagia therapy groups. Other subgroup analyses were conducted to compare effect sizes between selected interventions (i.e., Shaker exercise, Chin Tuck Against Resistance exercise [CTAR], and Expiratory Muscle Strength Training [EMST]), medical diagnoses, and outcome measures. Only between-subgroup meta-analyses were conducted using post-intervention data, to account for possible spontaneous recovery during the period of intervention.

Using Comprehensive Data Analysis software, publication bias was assessed following the Begg and Muzumdar’s rank correlation test and the fail-safe N test. The Begg and Muzumdar’s rank correlation test reports the rank correlation between the standardised effect size and the variances of these effects [17]. This statistical procedure produces tau as well as a two tailed *p* value; values of zero indicate no relationship, whereas deviations away from zero indicate a relationship. High standard error would be associated with larger effect sizes if asymmetry is caused by publication bias. Tau would be positive if larger effects are presented by low values, while tau would be negative if larger effects are represented by high values.

The fail-safe N test calculates how many studies with effect size zero could be added to the meta-analysis before the result lost statistical significance. That is, the number of missing studies that would be required to nullify the effect [18]. If this number is relatively small, then there is cause for concern. However, if this number is large, it can be stated with confidence that the treatment effect, while possibly inflated by the exclusion of some studies, is not nil.

## 3. Results

### 3.1. Study Selection

A total of 8059 studies were retrieved across four databases: CINAHL (*n* = 239), Embase (*n* = 4550), PsycINFO (*n* = 231), and PubMed (*n* = 3039). After removal of duplicate titles and abstracts (*n* = 1113), a total of 6946 records remained. After assessing titles and abstracts, 261 original articles were identified. Full-text records were accessed to verify all inclusion criteria. During full-text assessment, articles were divided into different types of interventions, as this systematic review reports on behavioural interventions only. Based on the inclusion criteria, 36 articles were included, after which one study was identified through reference checking of the included articles. Figure 1 presents the flow diagram of the article selection process according to PRISMA.

### 3.2. Description of Studies

All 37 included studies are described in detail in Table 2 and Table 3. Table 2 reports on study characteristics, definitions and methods of diagnosing oropharyngeal dysphagia, and details on participant groups. Information such as medical diagnosis, sample size, age and gender, is provided on all study groups. Table 3 presents intervention goals, intervention components, outcome measures and treatment outcome of each included study.

*Participants* (Table 2). The 37 studies included a total of 2656 participants (mean = 72; SD = 124.5), with the sample sizes across studies ranging from 10 [30] to 742 participants [38]. All but two studies reported the mean age of participants [38,49], which was 65.6 years (SD = 8.8). Participant age range was only reported in five studies, ranging between 55 [36] and 95 [38] years. The mean percentage of male participants across all studies was 55.8% (SD = 13.7).

Most studies included stroke patients (*n* = 24). Other diagnoses included: patients with Parkinson’s disease [19,39,52], acquired brain injury [30], multiple sclerosis [51] and nasopharyngeal cancer [50]. Two studies included a mixed patient population with Parkinson’s disease or dementia [38], and stroke or head and neck cancer patients after chemoradiation [40]. Five studies did not provide further details on diagnoses [28,38,49,54,55]. The most frequent method for confirming OD was VFSS (*n* = 17), with only four studies using FEES (*n* = 4) [20,31,38,40]. Seven studies used non-instrumental clinical assessments, five studies used a screening tool [28,29,39,48,56], and four studies used patient self-reported dysphagia [49,52,54,55]. The included studies were conducted across fifteen countries, with studies most frequently conducted in Korea (*n* = 13), USA (*n* = 6), China (*n* = 3) and Japan (*n* = 3).

*Outcome measures* (Table 3). Many different outcome measures were used across the included studies targeting different domains within the area of OD. The most frequently used measures were the Penetration Aspiration Scale (PAS; 15 studies), the Functional Oral Intake Scale (FOIS; 8 studies), various water swallow tests (4 studies), and the Mann Assessment of Swallowing Ability (MASA; 3 studies). All other outcome measures were used in one or two studies only, confirming the substantial heterogeneity in outcome measures.

*Interventions* (Table 3). The included 37 studies comprised a range of behavioural interventions, delivered by various health professionals. The interventions were most frequently implemented by single allied health disciplines: occupational therapists in ten studies, speech pathologists in eight studies, physical therapists in two studies [36,48], and nursing staff in one study [55]. In five studies, more than one discipline was involved [23,27,28,33,48], and two studies reported caregivers as the intervention agent either as a single agent [24] or in addition to occupational therapists [22]. Nine studies did not specify disciplines involved in providing the interventions. The intervention dosage varied greatly, ranging from one training session [54] to exercise 3 times daily, 7 days per week for 42 days [25].

*Behavioural intervention groups* (Table 3). Of the 37 included studies, seven studies comprised three participant groups [19,20,21,23,25,26,38], whereas all other studies included two groups. Based on authors’ description of therapy contents, all intervention groups were categorised into compensatory, rehabilitative, and combined compensatory and rehabilitative interventions. Ten studies included different types of intervention groups (i.e., compensatory, rehabilitative and/or combined compensatory and rehabilitative intervention groups). Five studies included only compensatory groups [20,24,38,39,55], ten studies included only rehabilitative groups, and thirteen studies included only groups combining compensatory and rehabilitative interventions.

Most studies (*n* = 23) included a comparison group that received a type of dysphagia treatment often referred to as traditional therapy, standard swallow therapy, or conventional dysphagia treatment (CDT). Some studies also used the term usual care for CDT groups. CDT treatment could include counselling and the provision of information about swallowing and dysphagia, compensatory strategies (e.g., bolus modification and adjusted head positioning), rehabilitation, oromotor exercises and/or thermal stimulation. Three studies included a comparison group receiving medical standard care without dysphagia treatment [20,51,56]. In three studies, patients underwent sham dysphagia training [36,43,53]. Several studies compared two or three behavioural interventions without having a CDT or medical standard care group included [33,34,46,49,50,55].

### 3.3. Risk of Bias Assessment

The Begg and Mazumdar rank correlation procedure produced a tau of 0.305 (two-tailed *p* = 0.113), indicating there is no evidence of publication bias. This meta-analysis incorporates data from 15 studies, which yield a *z*-value of 7.528 (two-tailed *p* < 0.001). The fail-safe N is 207. This means that 207 ‘null’ studies need to be located and included for the combined two-tailed *p*-value to exceed 0.050. That is, there would need to be 13.8 missing studies for every observed study for the effect to be nullified. Both of these procedures (i.e., Begg and Mazumdar rank correlation and fail-safe N) indicate the absence of publication bias.

### 3.4. Methodological Quality

Risk of bias of the included RCTs was assessed using the RoB 2 tool. Figure 2 and Figure 3 present the risk of bias summary per domain for individual studies and for all included studies. Most studies showed low risk of bias per domain, but more than half of the included studies (19/37) scored overall as having some concerns, with three studies identified as being at high risk.

### 3.5. Meta—Analysis: Effect of Interventions

Twenty-one studies were included in the meta-analyses [21,22,24,25,28,29,30,31,34,35,40,41,42,43,44,45,46,49,51,52,54]. All study groups were categorised into compensatory interventions, rehabilitative interventions, combined compensatory and rehabilitative interventions, and no dysphagia intervention. Seventeen studies were excluded from meta-analyses: one study included patients with self-reported swallowing difficulties without confirmed OD diagnosis by instrumental assessment (VFSS or FEES) [48], four studies did not report on instrumental or clinical non-instrumental outcome data [20,28,37,40], ten studies provided insufficient data for meta-analysis [21,24,27,34,38,39,48,51,56,57], and two studies were excluded to reduce heterogeneity between studies [32,53].

*Overall, within group analysis*. (Figure 4). A significant, large pre-post intervention effect size was calculated using a random-effects model (*z(35)* = 8.047, *p* < 0.001, Hedges’ *g* = 1.139, and 95% CI = 0.862–1.416). Pre-post intervention effects varied greatly between studies, ranging from 0.058 to 5.732. Of the 36 intervention groups included in the meta-analysis, 19 groups showed large effect sizes (Hedges’ *g* > 0.8), six groups showed moderate effects sizes (0.5 < Hedges’ *g* ≤ 0.8), seven groups showed minor effect sizes (0.2 < Hedges’ *g* ≤ 0.5), and four groups showed negligible effect sizes (Hedges’ *g* ≤ 0.2). Between-study heterogeneity was significant (*Q*(35) = 152.938, and *p* < 0.001), with *I^2^* showing heterogeneity accounted for 77.115% of variation in effect sizes across studies.

*Between subgroup analyses*. Subgroup analyses (Table 4) were conducted comparing different types of interventions: behavioural interventions were compared with conventional dysphagia treatment (CDT), or no dysphagia therapy groups (Figure 5). Both behavioural interventions and CDT were categorised into mainly compensatory, rehabilitative, and combined compensatory and rehabilitative interventions. Overall, significant treatment effects were identified favouring behavioural interventions. In particular, large effect sizes were found when comparing rehabilitative interventions with no CDT, and combined interventions with compensatory CDT. When comparing selected interventions based on commonalities across studies against CDT, significant, large effect sizes were found in favour of Shaker exercise, chin tuck against resistance exercise (CTAR), and expiratory muscle strength training (EMST). Most studies were conducted in stroke populations and showed significant, moderate effect sizes. Comparisons between outcome measures indicated at significant effects for PAS only.

## 4. Discussion

This systematic review aimed to determine the effects of behavioural interventions in people with OD based on the highest level of evidence (RCTs) only. Findings from the literature were reported using PRISMA and meta-analysis procedures.

### 4.1. Systematic Review Findings

In total, 37 behavioural RCTs in OD were identified. Considering the high prevalence [3] and severe impact of OD on health [57], quality of life [5,58], and health-economics [59], the limited number of high-level evidence studies is concerning. RCTs are costly and usually require extensive funding [60]. Possibly, the general lack of awareness of OD [61] might place funding applications in this research area at a disadvantage when competing with well-known, life-threatening diseases such as cancer or stroke. Although OD is a symptom of these diseases, and many other underlying conditions, limited public knowledge persists, resulting in reduced understanding and recognition of the devastating consequences of OD, in both health-care and non-health-care practitioners [61].

Further, although RCTs are characterised by random allocation and allocation concealment, few of the included studies included sufficient reporting on the processes of randomization and blinding. These finding are in line with current literature on quality assessments of RCTs [62,63], confirming that the risk of selection bias [63] and the success of blinding methods in RCTs [62] can often not be ascertained due to frequent poor reporting.

When comparing behavioural RCTs in OD, several methodological challenges arise. Authors may use different definitions for OD or fail to provide sufficient details when reporting on the swallowing problems of the included patient populations. Also, several studies used non-instrumental assessments (i.e., patient self-report or a screening tool) to identify or confirm OD, making the comparison between studies precarious. The use of a screening tool is especially problematic in identifying OD and cannot act as confirmation of OD. A screening tool’s purpose is merely to identify patients at *risk* of OD, after which further assessment may confirm or refute the diagnosis [2]. Additionally, although instrumental assessment is considered the optimal tool for confirming OD diagnosis, VFSS and FEES protocols may differ (e.g., using different numbers of swallow trials, viscosities, and volumes).

Studies used a wide range of outcome measures to evaluate treatment effects. Since OD is a multidimensional phenomenon [64], different dimensions of OD may result in different therapy outcomes. For example, changes in dysphagia-related quality of life or oral intake do not necessarily correlate with findings from instrumental assessment. As such, to reduce heterogeneity in meta-analyses, patient self-report and oral intake measures were excluded. Also, some studies included outcome measures with poor or unknown psychometric properties, which in turn undermines the interpretation of treatment effects as data may not be valid or reliable. In addition, measures with weak responsiveness characteristics are not sensitive to treatment changes and should therefore be avoided as outcome measures aiming to determine intervention effects [2].

Most studies included a combined rehabilitative and compensatory intervention group or a rehabilitative intervention group, with only a few studies including exclusively compensatory groups. As the interventions classified as CDT comparison groups showed large variation as well, CDT comparison groups were categorised into similar group types (compensatory and/or rehabilitative CDT). Overall, terminology in the literature referring to CDT comparison groups was varied and complex. This was especially pertinent when interventions were not described in sufficient detail and descriptive terms such as “usual care” or “traditional therapy” did not provide further clarity on the type or content of CDT provided. Despite using categories to group different types of interventions, some degree of heterogeneity was inevitable. Interventions used different types of exercises or care, in distinct dosages, and were applied by different health care professionals. Therefore, it is challenging to identify the “active” ingredients of individual interventions, especially as most studies combined the use of different treatment strategies.

### 4.2. Meta-Analysis Findings

When considering meta-analyses for behavioural interventions, overall significant treatment effects were identified as favouring behavioural interventions over CDT and withholding dysphagia therapy. Most promising intervention approaches were rehabilitative interventions, which were associated with large effect sizes. Additionally, rehabilitative interventions such as Shaker exercise, CTAR exercise, and EMST showed significant, large effect sizes. However, since most studies included in the meta-analysis provided data on stroke patients only, future research still needs to confirm these findings in other diagnostic populations such as Parkinson’s disease, acquired brain injury or patients with head and neck oncology. As stated above, patient self-report and oral intake measures were excluded from meta-analyses to increase homogeneity between studies. Though self-report and oral intake data might be interesting for future meta-analyses, this would require additional RCTs to be published, as currently there is limited data available in the literature. Finally, future studies should report on treatment dosage and duration in more detail. Due to high heterogeneity between studies and incomplete reporting, no subgroup meta-analyses could be conducted for these variables.

### 4.3. Limitations

Although reporting of this review followed the PRISMA guidelines to reduce bias, some limitations are inherent to this study. As only RCTs published in English were included, some RCTs may have been excluded based on language criteria. In addition, meta-analyses were restricted because of heterogeneity of the included studies. As such, comparisons across studies are challenging and, generalisations and meta-analyses results should be interpreted with caution.

## 5. Conclusions

Meta-analyses for behavioural studies in oropharyngeal dysphagia identified an overall, significant, large pre-post interventions effect size. Significant treatment effects were identified favouring behavioural interventions over conventional dysphagia treatment. Notably, large effect sizes were found when comparing rehabilitative interventions with no dysphagia treatment and combined interventions with compensatory conventional dysphagia treatment. Selected interventions compared with conventional dysphagia treatment showed significant, large effect sizes in favour of Shaker exercise, CTAR, and EMST.

Behavioural interventions show promising effects in people with oropharyngeal dysphagia. Still, generalisations from this meta-analysis need to be interpreted with care due to high heterogeneity across studies.

## Figures and Tables

**Figure 1 jcm-11-00685-f001:**
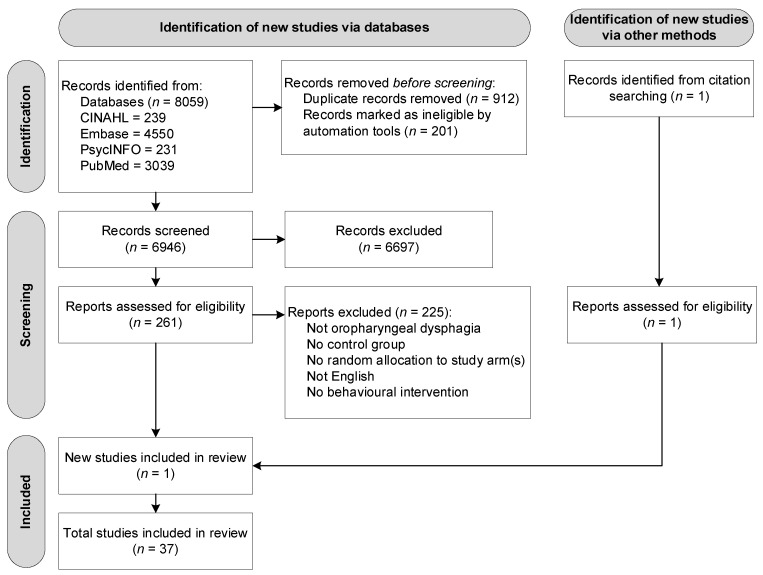
Flow diagram of the reviewing process according to PRISMA.

**Figure 2 jcm-11-00685-f002:**
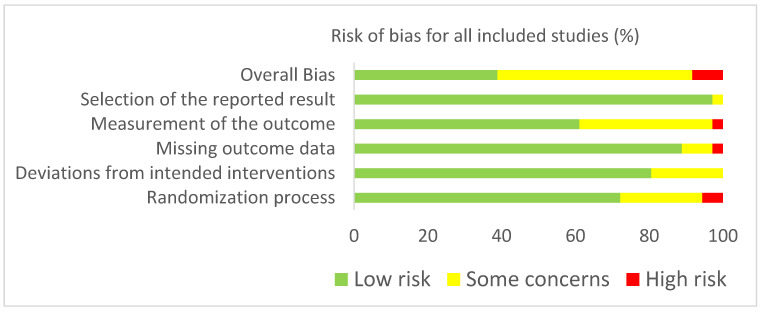
Risk of bias summary for all included studies (*n* = 37) in accordance with RoB2.

**Figure 3 jcm-11-00685-f003:**
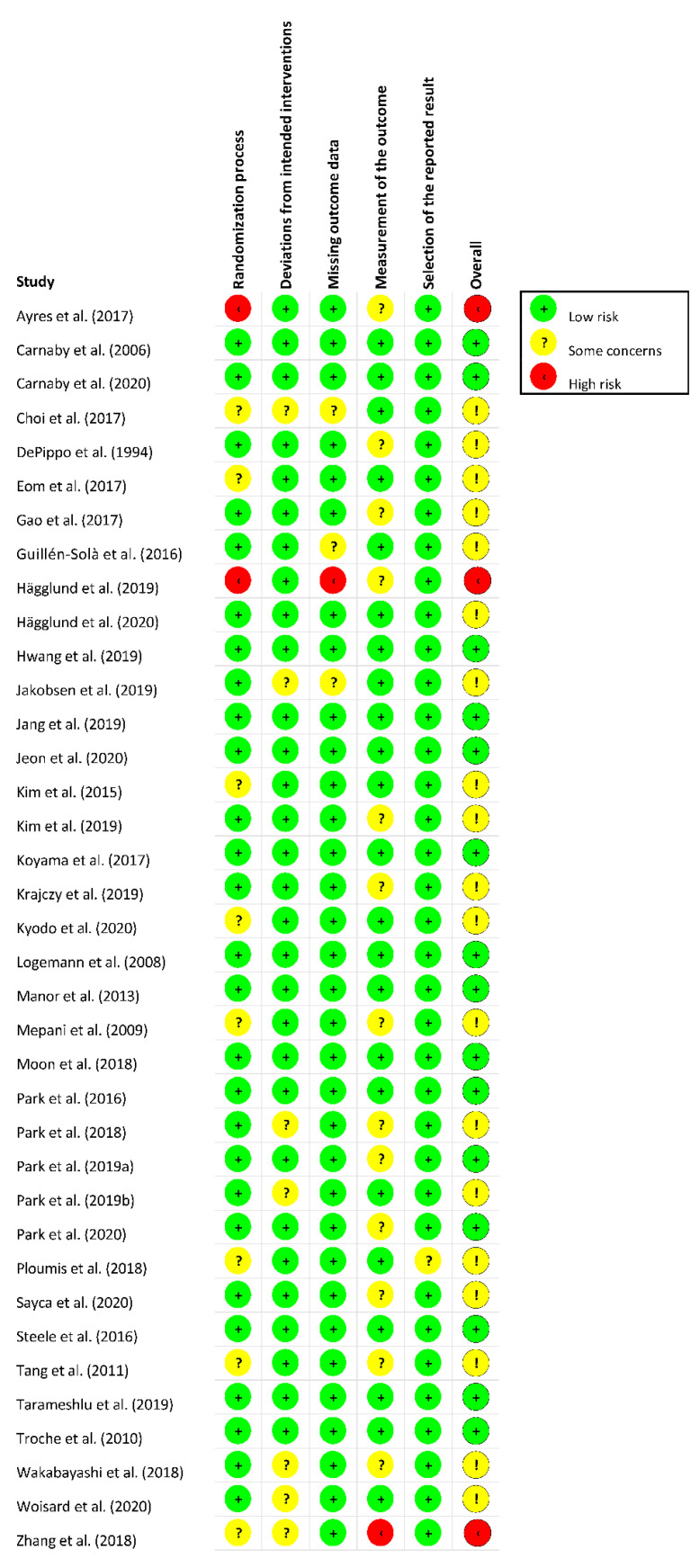
Risk of bias summary for individual studies (*n* = 37) in accordance with RoB2 [19,20,21,22,23,24,25,26,27,28,29,30,31,32,33,34,35,36,37,38,39,40,41,42,43,44,45,46,47,48,49,50,51,52,53,54,55]. *Note*. If one or more yellow or red circles (domains) have been identified for a particular study, the Overall score (last column) shows an exclamation mark, indicating that either the study shows some concerns (yellow circle with exclamation mark) or is at high risk (red circle with exclamation mark).

**Figure 4 jcm-11-00685-f004:**
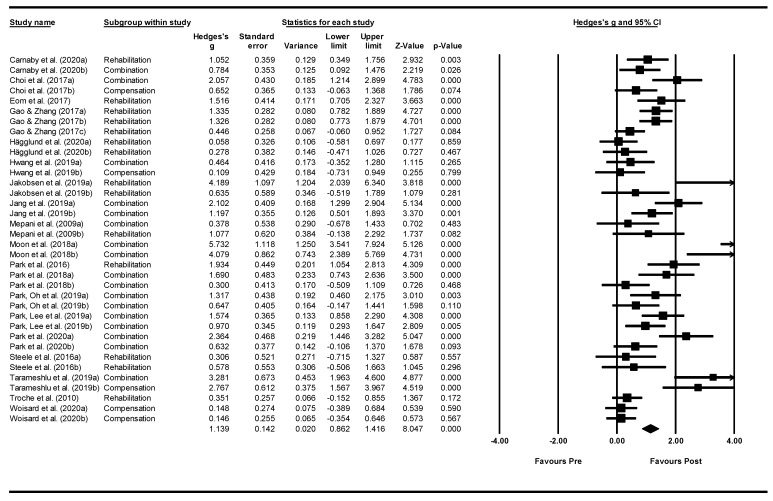
Within intervention group pre-post meta-analysis [21,22,24,25,28,29,30,31,40,41,42,43,44,45,46,49,51,52,54,56]. *Note.* Refer to Table 2 for explanation of the subgroups.

**Figure 5 jcm-11-00685-f005:**
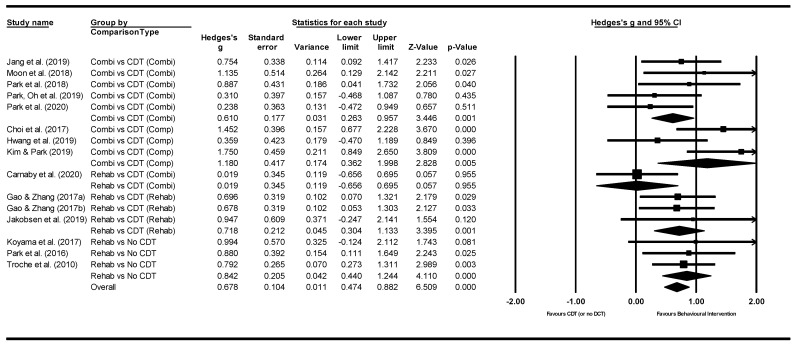
Between subgroup meta-analysis for different types of interventions: behavioural interventions compared with conventional dysphagia treatment (CDT) or no dysphagia therapy [21,22,25,29,30,31,34,35,41,42,43,44,46,52]. *Note.* Refer to Table 2 for explanation of the subgroups.

**Table 1 jcm-11-00685-t001:** Search strategies.

Database and Search Terms	Number of Records
Cinahl: ((MH “Deglutition”) OR (MH “Deglutition Disorders”)) AND (MH “Randomized Controlled Trials”)	239
Embase: (swallowing/OR dysphagia/) AND (randomization/or randomized controlled trial/OR “randomized controlled trial (topic)”/OR controlled clinical trial/)	4550
PsycINFO: (swallowing/OR dysphagia/) AND (RCT OR (Randomised AND Controlled AND Trial) OR (Randomized AND Clinical AND Trial) OR (Randomised AND Clinical AND Trial) OR (Controlled AND Clinical AND Trial)).af.	231
PubMed: (“Deglutition” [Mesh] OR “Deglutition Disorders” [Mesh]) AND (“Randomized Controlled Trial” [Publication Type] OR “Randomized Controlled Trials as Topic” [Mesh] OR “Controlled Clinical Trial” [Publication Type] OR “Pragmatic Clinical Trials as Topic” [Mesh])	3039

**Table 2 jcm-11-00685-t002:** Study characteristics of studies on behavioural interventions for people with oropharyngeal dysphagia.

Study Country	OD (Definition/Terminology; Diagnostic Measure/Method) Diagnosis Main Inclusion/Exclusion Criteria	Sample (N) Groups (*n*) ^a^	Group Descriptive (Mean ± SD) (Age, Gender, Relevant Medical Diagnoses)
Ayres, et al. [19] Brazil	*OD:* Oropharyngeal dysphagia determined by FEES *Diagnosis:* PD *Inclusion:* PD and oro-pharyngeal dysphagia. *Exclusion:* Presenting language and/or hearing disorders that could complicate the understanding of intervention; diagnosis of dementia, or other neurological illnesses.	*n* = 32: - Experimental group: Chin-down manoeuvre and swallowing orientation (*n* = 11) - Orientation group: Swallowing orientation only (*n* = 7) - Control group: No intervention (*n* = 14)	Experimental group/Orientation group/control group *Age years:* 62 (11.5)/64.5 (5.6)/62.8 (6.2) *Male:* 80%/66.7%/75% *Schooling*: 5.9 (4.1)/12 (9.1)/10.3 (8.4) *Time of disease:* 10.7 (4.7)/11.8 (8)/8.8 (6) *H & Y disability score:* 2.8 (0.8)/2.5 (0.7)/2.5 (0.8) *MOCA*: 21.9 (4.9)/20.5 (7.7)/21.2 (8.4) *PDQ-39:* 41.4 (13.8)/38.7 (16.7)/36.5 (17.1) *BDI:* 13.8 (7.7)/17.1 (9.2)/14.7 (9.3) *FOIS:* 5.9 (1.3)/6.8 (0.5)/6.8 (0.4)
Carnaby, et al. [20] USA	*OD:* Diagnosis of swallowing difficulty by speech pathologist, <85 on Hospital’s dysphagia assessment *Diagnosis:* Clinician diagnosed Stroke, WHO definition *Inclusion:* Stroke < 7 days *Exclusion: NR*	*n* = 306: - UC (*n* = 102) - Low intensity (*n* = 102) - High intensity (*n* = 102)	High intensity/low intensity/UC: mean (SD) *Age yr:* 69.8 (12.5)/72 (12.4)/71.4 (12.7) *Male:* 59%/58%/58% *Severity Barthel index <15:* 78%/78%/79% *Rankin score >3:* 85%/79%/83% *Length hospital stay, days:* 19.1/19.2/21.4
Carnaby, et al. [21] USA	*OD:* Dysphagia on admission- score < 178 on MASA, no history of swallowing disability, head/neck surgery. *Diagnosis:* Sub-acute stroke confirmed by attending neurologist according to the WHO definition *Inclusion:* Able to adhere to behavioural treatment regimens *Exclusion:* NR	*n* = 53: - MDTP + NMES (NMES; *n* = 18), - MDTP + sham NMES (MDTP; *n* = 18) [Denoted as ‘Carnaby et al. (2020a)’ in Figure 4.] - UC (*n* = 17) [Denoted as ‘Carnaby et al. (2020b)’ in Figure 4.]	NMES/MDTP/UC: mean (SD) *Age yr:* 62.7 (12.2)/70.6 (11.8)/64.3 (14.7) *Male:* 55%/44%/41% *Modified Rankin:* 4.5 (0.6)/4.46 (0.5)/4.56 (0.5) *Modified Barthel:* 5.3 (3.4)/5.5 (2.8)/5.6 (2.6) *Days post stroke:* 7.83 (3.9)/8.47 (7.17)/6.7 (5.1) *MASA score:* 157.8 (16.5)/154.62 (18.87)/158.4 *FOIS score:* 3.72 (1.44)/3.25 (1.61)/4.35 (1.8)
Choi, et al. [22] Korea	*OD:* Dysphagia after stroke confirmed by VFSS *Diagnosis:* Stroke (method NR) *Inclusion:* No major cognitive deficit (MMSE >20), >fair grade on neck muscle testing, symmetric neck posture *Exclusion:* neck pain or neck surgery, poor general condition, severe communication problem, unstable medical condition, presence of a tracheostomy tube	*n* = 32: - Experimental–Shaker exercise (SE) and conventional dysphagia therapy (CDT; *n* = 16) [Denoted as ‘Choi et al. (2017a)’ in Figure 4.] - Control–CDT (*n* = 16) [Denoted as ‘Choi et al. (2017b)’ in Figure 4.]	Experimental SE + CDT/control (CDT): mean (SD) *Age yr:* 60.8 (10.9)/60.4 (10.5) *Gender (male/female):* 10/6/9/6 *Time since stroke onset months:* 3.4 (1.6)/4.1 (1.0) *PAS:* 4.6 (0.8)/4.9 (0.1) *FOIS:* 3.1 (1.0)/3.2 (0.68)
DePippo, et al. [23] USA	*OD:* MBS, BDST, VFSS, speech pathologists determined dysphagia *Diagnosis:* Stroke by clinical history, neurologic examination CT/MRI *Inclusion:* 20–90 yrs, no history of oral or pharyngeal anomaly *Exclusion:* aspirated >50% of all consistencies,	*n* = 115, allocated to graded therapist treatment levels: - Group A (*n* = 38) - Group B (*n* = 38) - Group C (*n* = 39)	Group A/Group B/Group C *Age yr:* 76/74.5/73 *Male/Female:* 22/16/19/19/27/12 *Mini-Mental State score:* 16 (12)/17 (10)/18 (10) *Barthel-ADL Mobility:* 37 (23)/48 (20)/46 (38) *Weeks post stroke:* 4.6/4.5/4.9
Eom, et al. [24] Korea	*OD: D*ysphagia caused by a stroke, confirmed by VFSS *Diagnosis:* Stroke *Inclusion:* Age > 65, onset duration < 3 months, score ≥ 24 on MMSE. *Exclusion:* Presence of severe orofacial pain, significant malocclusion or facial asymmetry, unstable breathing or pulse, tracheostomy, aphasia or apraxia, inadequate lip closure	*n* = 30: - Experimental- resistance expiratory muscle strength training (*n* = 15) - Placebo group (*n* = 15).	Experimental/Placebo *Age yr:* 69.2 (4.1)/70.2 (3.6) *Male/Female:* 5/8/6/7 PAS baseline: 5.1 (0.8)/4.9 (0.6)
Gao and Zhang [25] China	*OD:* VFSS evaluation *Diagnosis:* Chinese diagnosis guidelines for acute ischemic stroke, CT or MRI *Inclusion:* >60 yrs, positive Neill screening test, first-time cerebral infaction. *Exclusion:* unstable conditions, previous abnormality in mouth, throat or neck, multiple organ dysfunction syndromes, uncooperative patients, severe mental illness, complete or sensory aphasia	*n* = 90: - Control (*n* = 30) [Denoted as ‘Goa & Zhang (2017c)’ in Figure 4.] - Shaker exercise (*n* = 30 [Denoted as ‘Goa & Zhang (2017a)’ in Figure 4.] - Chin tuck against resistance (CTAR; *n* = 30) [Denoted as ‘Goa & Zhang (2017b)’ in Figure 4.] - [Figure 5: Shaker–Control denoted as ‘Goa & Zhang (2017b)’; CTAR–Control: denoted as ‘Goa & Zhang (2017a)]	Control/Shaker/CTAR *Age yr:* 71.1 (6.4)/71.1 (7.1)/70.9 (6.6) *Male:* 14/15/13 *Therapeutic course (day):* 12.2 (1.4)/13.0 (1.4)/ 13.0 (1.6)
Guillén-Solà, et al. [26] Spain	*OD:* Dysphagia confirmed by VFSS score ⩾3 in 8-point PAS *Diagnosis:* Subacute ischemic stroke *Inclusion:* Stroke within 1–3 wks. *Exclusion:* Cognitive impairment and/or history of previous neurological diseases associated with dysphagia	*n* = 62: - Group 1: control standard swallow therapy (SST) (*n*= 21), - Group 2: SST + IEMT (*n* = 21) - Group 3: SST + sham IEMT+ NMES (*n* = 20).	Control/IEMT/NMES *Age yr:* 68.9 (7.0)/67.9 (10.6)/70.3 (8.4) *Male:* 12 (57.1%)/16 (76.2%)/10 (47.6%) *Modified Rankin:* 3.7 (0.8)/3.9 (0.5)/3.6 (0.8) *Barthel Index:* 44.0 (18.5)/42.7 (14.6)/41.8 (12.2) *Stroke onset (days):* 9.3 (5.1)/10.8 (8.7)/11.0 (5.5) *FOIS*: 4.3 (0.6)/4.5 (0.5)/4.4 (1.0) *PAS:* 5.4 (2.3)/5 (2.7)/5.5 (2.2)
Hägglund, et al. [27] Sweden	*OD:* Swallowing function assessed with timed water swallow test; diagnosed dysfunction when swallowing rate did not exceed 10 mL/s. *Diagnosis:* NR *Inclusion:* ≥65 yrs, No cognitive impairment, ≥3 days intermediate care. *Exclusion:* Patients receiving end of life care, moderate or severe cognitive impairment	*n* = 116: - Intervention: Oral neuromuscular training (*n* = 49) - Control: Usual care (*n* = 67)	Control/Intervention *Age yr:* 85/83 *Male:* 29 (43.3)/27 (55.1) *Dysphagia risk condition:* 32 (47.8)/25 (52.1) *Care moderate dependence:* 27 (40.9)/18 (36.7) *Swallowing rate (mL/s):* 4.10/5.31
Hägglund, et al. [28] Sweden	*OD:* Swallowing dysfunction (pathological TWST test 4-weeks post-stroke) *Diagnosis:* Stroke *Inclusion:* First-time stroke and a pathological timed water swallow test. *Exclusion:* Inability to cooperate; neurological diseases other than stroke, known history of dysphagia prior to the stroke, prominent horizontal overbite (contra-indication due to the oral device’s design), or hypersensitivity to the acrylate	*n*= 40: - Control group: 5 weeks of continued of oro-facial sensory stimulation (*n* = 20) [Denoted as ‘Hägglund et al. (2020b)’ in Figure 4.] - Intervention group: Oral neuromuscular training using oral device (Muppy^®^) for 5 weeks + oro-facial sensory vibration stimulation (*n* = 20) [Denoted as ‘Hägglund et al. (2020a)’ in Figure 4.]	Control/Intervention *Age years:* 75 (56–90) yrs/75 (60–85) Male = 14/11; Female: 6/9. *Stroke type:* Ischemic (16 16); ICH= 3/4; ischemic and ICH = 1/0; left hemisphere = −6/7; right hemisphere = 10/10; supratentorial = 15/16; infratentorial = 3/4; supra-and infratentorial = 1/0. *Lowered consciousness at hospital admission:* 6/6
Hwang, et al. [29] Korea	*OD:* OD confirmed by VFSS *Diagnosis:* Stroke *Inclusion:* Dysphagia *<*3 months, swallow voluntarily. Exclusion: trigeminal neuropathy, tongue deviation, facial asymmetry, communication disorders.	*n* = 25: - Experimental, tongue stretching exercises (TSE) (*n* = 13) [Denoted as ‘Hwang et al. (2019a)’ in Figure 4.] - Control group (*n* = 12) [Denoted as ‘Hwang et al. (2019b)’ in Figure 4.]	Experimental/Control *Age (yrs):* 60.5 (12.5)/62.2 (10.3) *Male:* 6/5 *Time since stroke, weeks:* 8.2 (2.9)/9.1 (2.7) *Type of stroke (n) Haemorrhage:* 7/6 *Type of stroke (n) Infarction:* 4/4
Jakobsen, et al. [30] Denmark	*OD:* Clinical signs of dysphagia; score ≥3 on PAS, FEES. *Diagnosis:* Severe ABI, non-sedated GCS <9, <24 hrs of injury *Inclusion:* 18–65 yrs *Exclusion:* formerly acquired or congenital brain damage, psychiatric diagnosis, history of treatment for head and neck cancer, need for a tracheostomy tube, agitated behaviour	*n* = 10: - Intervention facilitation of swallowing (*n* = 5) [Denoted as ‘Jakobsen et al. (2019a)’ in Figure 4.] - Control basic care + usual treatment (*n* = 5) [Denoted as ‘Jakobsen et al. (2019b)’ in Figure 4.]	Control/Intervention *Age yrs:* 45.6 (37.5–57.8)/53.8 (41.8–61.4) *Male:* 4/2 *Days from injury:* 70.4 (43.0)/76.4 (21.8) *GCS at injury* (3–15 points): 6.8 (4.4)/6.0 (5.2)
Jang, et al. [31] Korea	*OD:* Swallowing difficulty VFSS-patients who showed velopharyngeal incompetence (VPI) on VFSS were enrolled *Diagnosis:* Subacute stroke *Inclusion:* Diagnosis of subacute stroke *Exclusion:* Previous stroke, pharyngeal structural abnormalities, unable to cooperate	*n* = 36: - Study-conventional therapy + mechanical inspiration, expiration exercise (*n* = 18) [Denoted as ‘Jang et al. (2019a)’ in Figure 4.] - Control-conventional therapy only (n = 18) [Denoted as ‘Jang et al. (2019b)’ in Figure 4.]	Study/Control *Age yrs:* 67.3 (9.5)/71.15 (8.6) *Male, n:* 10/9 *Stroke type, n Haemorrhage:* 8/6 *Days from stroke onset:* 20.5 (13.6)/18.4 (12.5)
Jeon, et al. [32] Korea	*OD:* Swallowing dysfunction/dysphagia as determined by VDS and PAS scores on VFSS *Diagnosis:* Stroke disease *Inclusion:* MMSE-K score ≥19 points; stroke disease duration ≥6 mths and <2 years Exclusion: Altered neck posture. VitalStim contraindications or cardiopulmonary disease.	*n*= 34: - Experimental group: NMES + upper cervical spine mobilisation (*n* = 17) - Control group: NMES and sham mobilization (*n* = 17)	Experimental/Control *Age yrs:* 63.12 (13.5)/64.47 (8.43) *Male*: 11/6; 11/6 *Side of stroke (left/right):* 6/11; 7/10 *Haemorrhage/infarction*: 14/3/12/5 *Weight:* 69.11 (11.95); 65.55 (12.66) K-MMSE (point): 24.53 (2.62)/24.2 (2.91) K-NIHSS (point): 10.41 (3.06)/10.76 (3.75)
Kim, et al. [33] Korea	*OD:* Dysphagia defined as a disorder that causes difficulty with chewing and swallowing food *Diagnosis:* Stroke *Inclusion:* Diagnosed with dysphagia between May and July 2014; Symptoms of dysphagia for 6 months prior to treatment; 24 points or higher on MMSE- K; fair grade of manual muscle testing of neck flexors. Exclusion: Heart/internal/musculoskeletal disease	*n*= 26: - Experimental group: PNF short-flexion neck exercises (*n*= 13) - Control group: Shaker exercise (*n*= 13)	Experimental/Control *Age yrs:* 63.2 (10.2)/63.6 (8.1) *Male*: 8/5; 7/8 *Side of stroke (right/left):* 7/6/7/6
Kim and Park [34] Korea	*OD:* Dysphagia confirmed by VFSS *Diagnosis:* Diagnosed as having had stroke within 6 months post-onset *Inclusion:* Liquid aspiration or penetration on VFSS, nasogastric tube able to communicate, no cognitive deficit *Exclusion:* Secondary stroke, gastronomy tube, tracheostomy, neck or shoulder pain, cervical herniated nucleus, cervical spine orthosis or brainstem stroke	*n* = 30: - Experimental group, mCTAR exercise and traditional dysphagia treatment (*n* = 12) - Control group, only traditional (*n* = 13)	Experimental/Control *Age yrs:* 63.5 (5.5)/65.2 (6.2) *Male*: 6/6 *Type of stroke–haemorrhage*: 5/7 *Side of stroke (right/left):* 5/7/4/9 *Facial palsy:* 1/1 *Dysarthria:* 1/0
Koyama, et al. [35] Japan	*OD:* Stroke related dysphagia, hypopharyngeal residue found by VFSS *Diagnosis:* Stroke *Inclusion:* able to perform real or sham exercise *Exclusion*: Level 1 to 4 on FOIS, pulmonary aspiration with 2 mL of barium water in VFSS, past or present temporomandibular joint disease and/or tumor in head or neck, past or present progressive disease	*n* = 12: - Intervention, modified jaw opening exercise (MJOE; *n* = 6) - Control, isometric jaw closing exercise (*n* = 6)	Intervention/control *Age yrs:* 66.0 (9.3)/71.8 (7.6) *Male*: 5/5 *Post-onset weeks, mean (SD):* 6.7 (2.1)/9.2 (4.0) *FOIS, n, Level 5/Level 6:* 3/3/4/2
Krajczy, et al. [36] Poland	*OD:* Level 1–3 or 5–7 on SRS *Diagnosis*: Ischaemic stroke- using the National Institutes of Health Stroke Scale *Inclusion:* Early post-stroke (first stroke) period (<30 days) *Exclusion:* 2nd or 3rd stroke, level 1–3 dysphagia or level 5–7 dysphagia according to SRS, cognitive function disorders, total aphasia, anarthria, bilateral facial nerve paralysis, tracheostomy	*n* = 60: - Study, original dysphagia treatment (*n* = 30) - Control (*n* = 30)	Study/Control *Age yrs:* 55–65 (3.3)/55–65 (1.5) *Male:* 12/14 *Paresis, right side:* 15/12
Kyodo, et al. [37] Japan	*OD:* Dysphagia determined by endoscopic swallowing evaluation *Diagnosis*: Elderly patients with moderate-to-severe dysphagia. Diagnosis: NR *Inclusion:* Patients hospitalized between May 2017 and Sept 2018 who underwent endoscopic swallowing evaluation *Exclusion:* Patients ≥65 years old; the presence of an acute infection; patients who developed cerebrovascular disease, myocardial infarction, aspiration pneumonia within 2 weeks	*n*= 62 (randomized crossover trial): - Control group: Pureed diet without gelling agent - Intervention group: Pureed diet with gelling agent	Total sample *Age years:* 83 (9) *Male/female:* 36/26 Height (cm): 153.4 (6) Weight (kg): 51.8 (5) Concurrent medical conditions: - Aspiration pneumonia: 22 (35%) - CVA: 19 (31%) - Other: 21 (34%) *Hyodo-Komagane Score* Mild 0–3: 8 (13%) Moderate 4–7: 35 (56%) Severe 8–9: 19 (31%)
Logemann, et al. [38] USA	*OD:* Speech pathologist referral after swallow screening, patient aspirating thin liquids. *Diagnosis*: Physician’s diagnosis of dementia or PD. Bedford Alzheimer Nursing Severity Scale; neurologist rated PD using Hoehn and Yahr scale. *Inclusion:* 50–95 yrs *Exclusion:* Inability to perform chin down intervention	*n* = 742 - All patients received all 3 interventions (random order): - Chin-down intervention - Nectar - Honey-thickened liquids	*Age range:* 50–79, 41% *Age range:* 80–95, 59% *Male:* 70% *PD–No dementia:* 32% *PD–Dementia:* 19% *Dementia–Other:* 19% *Dementia–Single or multistroke*: 15% *Dementia–Alzheimer’s:* 15%
Manor, et al. [39] UK	*OD:* Referred to speech pathologist for evaluation of swallowing disturbances, confirmed via FEES. *Diagnosis*: PD had been diagnosed according to the UK Brain Bank criteria *Inclusion:* Diagnosis as above *Exclusion:* History of other uncontrolled neurological or medical disorders interfering with swallowing	*n* = 42: - Experimental group -received video-assisted swallowing therapy (VAST; *n* = 21) - Control group-conventional therapy (*n* =21)	Vast/Conventional therapy *Age yrs:* 67.7 (8.3)/69.9 (9.7) *Disease duration (years)* 7.4 (4.7)/8.8 (5.7) *Disease severity (H&Y-1–5)* 2.2 (0.8)/2.2 (0.8) *MMSE score (range 0–30)* 28.1 (1.6)/27.8 (1.5) *Swallowing disturbances questionnaire:* 14.7 (5.8)/14.3 (7.2) *Fiberoptic endoscopic evaluation of swallowing:* 0.7 (0.4)/0.6 (0.4)
Mepani, et al. [40] USA	*OD:* Post deglutitive dysphagia, pharyngeal phase dysphagia, VFSS to confirm *Diagnosis*: Stroke or chemoradiation for head and neck cancer *Inclusion:* Pharyngeal phase dysphagia, incomplete UES opening and postdeglutitive aspiration, hypopharyngeal residue, able to comply with protocol, dysphagia with aspiration of at least 3 month duration *Exclusion:* History of pharyngeal surgical procedures excluded.	*n* = 11: - Traditional swallowing therapy (*n* = 6) [Denoted as ‘Mepani et al. (2009a)’ in Figure 4.] - Shaker Exercise (*n* = 5) [Denoted as ‘Mepani et al. (2009b)’ in Figure 4.]	Traditional/Shaker *Age years:* 70.5 (9.5)/64 (22.8) *Males:* 5 (83%)/3 (60%) *Etiology of dysphagia:* - CVA: 4 (67%) 2 (40%) - Cancer: 2 (33%) 3 (60%)
Moon, et al. [41] Korea	*OD:* Aspiration or penetration, oropharyngeal residue, confirmed VFSS. *Diagnosis*: Subacute stage 3–12 weeks after the onset of stroke *Inclusion:* Diagnosis as above, could follow instructions provided, score of > 21 on Mini Mental State Exam, decreased lingual pressures with either anterior or posterior tongue as 40 kPa *Exclusion:* Nonstroke patients with dysphagia.	*n* = 16: - TPSAT plus traditional dysphagia therapy (*n* = 8) [Denoted as ‘Moon et al. (2018a)’ in Figure 4.] - Control, traditional dysphagia therapy (*n* = 8). [Denoted as ‘Moon et al. (2018b)’ in Figure 4.]	TPSAT/Control *Age years:* 62.0 (4.2)/63.5 (6.1) *Male:* 3/4 *Stroke type (ischemic/hemorrhagic):* 6/2/6/2 *Poststroke duration days:* 56.0 (17.4)/59.9 (20.0) *MMSE:* 22.87 ± 2.47 23.50 ± 2.00
Park, et al. [42] Korea	*OD:* Dysphagia confirmed by VFSS *Diagnosis*: Stroke *Inclusion:* Onset within 6 months; score ≥24 on the MMSE *Exclusion:* Stroke prior to that resulting in dysphagia, severe orofacial pain, significant malocclusion or facial asymmetry, unstable breathing or pulse, tracheostomy, severe communication disorder, inadequate lip closure	*n* = 27: - Experimental group, Expiratory muscle strength training (EMST) (*n* = 14) - Placebo sham (*n* = 13)	Experimental/Placebo *Age years:* 64.3 (10.7)/65,8 (11.3) *Male n:* 6/6 *Time since onset weeks:* 27. 4 (6.3)/26.6 (6.8)
Park, et al. [43] Korea	*OD:* Dysphagia following stroke was confirmed by VFSS *Diagnosis*: Stroke *Inclusion:* Onset duration was <12 months, swallow voluntarily, MMSE score ≥20 *Exclusion:* Secondary stroke, severe communication disorder, pain in the neck region, unstable medical conditions, head and neck cancer	*n* = 22: - Experimental, chin tuck against resistance exercise (CTAR; *n* = 11) [Denoted as ‘Park et al. (2018a)’ in Figure 4.] - Control group, only conventional dysphagia treatment (*n* = 11). [Denoted as ‘Park et al. (2018b)’ in Figure 4.]	Experimental/Control *Age years:* 62.2 (17.3)/58.4 (12.5) *Male:* 6/4 *Infarction:* 7/6 *Time after stroke (weeks):* 37.2 (54 3)/14 (14.4) *Oral feeding:* 4/5 *Tube feeding*: 7/6
Park, et al. [44] Korea	*OD:* OD after stroke by VFSS *Diagnosis*: Stroke based on computed tomography or MRI *Inclusion:* Inpatient, no significant cognitive problems (MMSE score *>* 24) *Exclusion:* Secondary stroke, trigeminal neuropathy, significant malocclusion or facial symmetry, parafunctional oral habits, tongue strength could not be measured, severe communication disorders, neck pain or neck surgery, presence of tracheostomy tube	*n* = 24: - Experimental, effortful swallowing training (EST; *n* = 12) [Denoted as ‘Park, Oh et al. (2019a)’ in Figure 4.] - Control, saliva swallowing (*n* = 12). [Denoted as ‘Park, Oh et al. (2019b)’ in Figure 4.]	Experimental/Control *Age years:* 66.5 (9.5)/64.8 (11.2) *Male:* 6/5 *Stroke lesion middle cerebral artery:* 6/6 *Time since stroke onset, wks:* 24.4 (8.6)/25.7 (6.3)
Park, et al. [45] Korea	*OD:* pharyngeal dysphagia confirmed through VFSS *Diagnosis*: Diagnosed as having stroke *Inclusion:* Within 6 months post-onset, nasogastric tube; absence of cognitive deficits. *Exclusion:* Secondary stroke, presence of other neurological, pain in the disc and cervical spine, cervical spine orthosis, presence of gastronomy tube, problems with the oesophageal phase of dysphagia	*n* = 37 patients: - Experimental, game-based chin tuck against resistance exercise (*n* = 19) [Denoted as ‘Park, Lee et al. (2019a)’ in Figure 4.] - Control, traditional head-lift exercise (*n* = 18) [Denoted as ‘Park, Lee et al. (2019b)’ in Figure 4.]	Experimental/Control *Age years:* 60.9 (11.2)/59.5 (9.3) *Male n:* 13/10 *Type of stroke, haemorrhage, n:* 12/14 *Paretic side, right, n:* 11/13 *Time since stroke, months*: 3.60 (1.19)/3.85 (1.18)
Park, et al. [46] Korea	*OD:* Dysphagia after stroke, by VFSS *Diagnosis*: Stroke due to hemorrhage or infarction *Inclusion:* <6 months of onset, liquid aspiration or penetration on VFSS; nasogastric tube; voluntary swallowing; coughing after water swallow test. *Exclusion:* Secondary stroke, difficulty in using both upper limbs, significant malocclusion or facial asymmetry, pain in the disc and cervical spine, limitations in opening jaw, use of cervical spine orthosis, tracheostomy, severe communication difficulties associated with dementia or aphasia, presence of gastronomy tube, problems with the oesophageal phase of dysphagia	*n* = 40: - Experimental, resistive jaw opening exercise (RJOE (*n* = 20) [Denoted as ‘Park et al. (2020a)’ in Figure 4.] - Placebo group (*n* = 20) [Denoted as ‘Park et al. (2020a)’ in Figure 4.]	Experimental/Placebo *Age years:* 62.1 (10.1)/61.8 (12.1) *Male:* 9/8 *Infarction:* 7/8
Ploumis, et al. [47] Greece	*OD:* Dysphagia screening-at least one severe symptom, validated in Greek Ohkuma questionnaire *Diagnosis*: Hemiparesis following stroke *Inclusion:* Hemiparesis following stroke, at least one severe symptom of the validated Greek Ohkuma questionnaire *Exclusion:* Exclusion-Barthel Index >20, Motor Function Hemispheric Stroke Scale <25, history of OD.	*n* = 70: - Experimental group cervical isometric exercises (*n* = 37) - Control (*n* = 33)	Experimental/Control *Age years (all participants):* 52 (15) *Barthel Index:* 22.8 (2.4)/23.4 (2.7) *Motor function, Stroke Scale:* 22.8 (2.4)/23.4 (2.7) *Sagittal C2-C7 Cobb angle:* 16.9 (18.5)/14.0 (16.2) *Coronal C2-C7 Cobb angle:* 6.9 ± 5.3/6.2 ± 5.0 *VFSS Score:* 1.0 (0)/1.0 (1.0)
Sayaca, et al. [48] Turkey	*OD: ‘*Swallowing difficulties’ determined with Turkish version of the eating assessment tool (T-EAT-10) *Diagnosis*: No neurological problems after neurologist’s examination *Inclusion:* Over 65 yrs, adequate cognitive status. Exclusion: Head/neck conditions affecting swallowing	*n* = 50: - Proprioceptive neuromuscular facilitation (PNF; *n* = 25) - Shaker exercises (*n* = 25)	Shaker/PNF *Age years:* 69 (4.9)/67 (2.1) *Male:* 10/10 *T-EAT-10 scores:* 3.5 (1.8)/3.6 (1.3) *Peak amplitude (μV):* 425.1 (170.7)/417.9 (143.0) *Swallow speed (secs):* 1.3 (0.3)/1.3 (0.3) *Swallow capacity (mL/sec):* 1.2 (0.1)/1.2 (0.1) *Swallow volume (mL/sec):* 1.3 (0.1)/1.3 (0.1)
Steele, et al. [49] Canada	*OD:* Dysphagia post stroke (VFSS) *Diagnosis*: Recent stroke (4–20 wks) *Inclusion:* Recent stroke, one repetition maximum posterior maximum isometric tongue-palate pressure measure <40 kPa at intake, stage transition duration if < 350 ms on at least one liquid barium swallow at intake VFSS *Exclusion:* Severe dysphagia with no functional opening of upper esophageal sphincter; pre-existing dysphagia or diagnoses of head and neck.	*n* = 14: - Experimental TPPT treatment arm (*n* = 7) [Denoted as ‘Steele et al. (2016a)’ in Figure 4.] - Comparison TPSAT treatment arm (*n* = 7) [Denoted as ‘Steele et al. (2016b)’ in Figure 4.]	TPPT/TPSAT *Age years, range:* 56–84/49–89 *Male*: 4/5 *Days post onset, range:* 28–126/33–150
Tang, et al. [50] China	*OD:* Radiation-induced dysphagia and trismus by non-instrumental clinical assessment *Diagnosis*: Nasopharyngeal carcinoma (NPC) patients after radiotherapy *Inclusion:* Diagnosed as above *Exclusion:* Dysphagia or trismus as initial symptoms of NPC excluded	*n* = 43: - Rehabilitation group, routine treatment + 3 months rehabilitation therapy (*n* = 22) - Control group, routine treatment (*n* = 21)	Rehabilitation group/Control group *Age years (total sample):* 49.3 (11) *Male (total sample), n:* 32 *Postradiotherapy, years:* 4.6 (1.8)/4.8 (1.6) *Interincisor distance (IID), cm:* 1.9 (0.7)/1.8 (0.6)
Tarameshlu, et al. [51] Iran	*OD:* Dysphagia based on DYMUS questionnaire (patient self-report) *Diagnosis*: Established diagnosis of MS according to McDonald’s criteria *Inclusion:* 20–60 years, lack of acute relapse in past two months, no other conditions such as stroke *Exclusion:* severe reflux, dysphagia due to drug toxicity, pregnancy	*n* = 20: - Experimental (TDT), sensorimotor exercises and swallowing manoeuvres (*n* = 10) [Denoted as ‘Tarameshlu et al. (2019a)’ in Figure 4.] - Usual Care (UC), diet prescription and postural changes (*n* = 10) [Denoted as ‘Tarameshlu et al. (2019b)’ in Figure 4.]	TDT/UC *Age years:* 47.5 (12.9)/39.9 (9.7) *Male:* 2/5 *Disease Duration (years):* 6.8 (2.9)/6.1 (2.7) *Expanded Disability Status Scale:* 3.6(2.1)/3.2(2.5) *MS Type-Relapse-Remitting:* 4/7 *MS Type-Primary Progressive*: 4/1 *MS Type-Secondary Progressive:* 2/2
Troche, et al. [52] USA	*OD:* Swallowing disturbance (screening followed by VFSS) *Diagnosis*: PD-diagnostic criteria of the UK Brain Bank *Inclusion:* 55–85 yrs, same PD medication, >24 MMSE. *Exclusion:* other neurologic disorders; head/neck cancer	*n* = 60: - Expiratory muscle strength training (EMST; *n* = 30) - Sham (*n* = 30)	EMST/Sham *Age years:* 66.7 (8.9)/68.5 (10.3) *Male:* 25/22 *Hoehn & Yahr stage 2.5:* 8/13, *stage 3*: 14/8 *Unified Parkinson’s Disease Rating Scale III motor total:* 39.4 (9.2)/40.0 (8.5)
Wakabayashi, et al. [53] Japan	*OD:* Dysphagia, Eating Assessment Tool (EAT-10) score ≥3 points *Diagnosis*: NR (Community-dwelling, ≥65 yrs) *Inclusion:* Receiving long-term care via day-service or day-care program, mild cognitive impairment/dementia *Exclusion:* Severe or moderate dementia, inability to perform training	*n* = 91: - Intervention group, resistance training of swallowing muscles (*n* = 43) - Control group (*n* = 48)	Intervention/Control *Age years:* 80 (7)/79 (7) *Male:* 19/28 *Tongue pressure (kPa):* 23.3 (8.3)/23.3 (10.0) *EAT-10, median (IQR):* 7 (5–13)/8 (4–11) *Barthel Index*: 81 (9)/81 (21)
Woisard, et al. [54] France	*OD:* Dysphagia- by Deglutition Handicap Index (DHI) *Diagnosis*: NR. (Sitting abnormality- by seated postural control measure, SPCM). *Inclusion* >18 years; DHI score >11, score >0 on 1 item SPCM, chronic dysphagia. *Exclusion:* NR	*n* = 56: - Group without device (D−) (*n* = 30) [Denoted as ‘Woisard et al. (2020b)’ in Figure 4.] - Group with the device (D+) (*n* = 26) [Denoted as ‘Woisard et al. (2020a)’ in Figure 4.]	D-/D+ *Age years (total sample):* 61.5 (11.8) *Male, n (total sample):* 35 *Degenerative dysphagia, N (total sample):* 24 *NIHSS:* 1.3 (1.4)/1.3 (1.6) *PAS:* 1.7 (1.3)/1.9 (1.9) *FOIS:* 6.0 (0.9)/5.8 (1.1)
Zhang and Ju [55] China	*OD:* Swallowing dysfunction (water swallow test upon inclusion) *Diagnosis*: Stroke *Inclusion:* Swallowing dysfunction *exclusion:* NR (admitted patients with dysphagia)	*n* = 120: - Intervention, nursing intervention (*n* = 60) - Control, conventional nursing service (*n* = 60)	Control/intervention *Age years:* 70.6 (7.4)/70.3 (7.4) *Males:* 33/32

^a^ Terminology as used by author(s). *Notes*. ABI = Acquired brain injury; BDI = Beck Depression Inventory; BDST = Burke Dysphagia Screening Test; CVA = cerebrovascular accident; DOSS = Dysphagia Outcome and Severity scale; FEES = Fiberoptic Endoscopic Evaluation of Swallowing; FOIS = Functional Oral Intake Scale; GCS = Glasgow Coma Scale; H&Y disability score = Hoehn and Yahr disability score; K-MMSE or MMSE-K = Mini-mental examination Korean version; K- NIHSS = Korean version of National Institute of Health Stroke Scale; MASA = Mann Assessment of Swallowing Ability; MBS = Modified Barium Swallow; MIE = Minimally Invasive Oesophagectomy; MDTP = McNeill Dysphagia Therapy Program; MMSE = Mini-Mental State Examination; MOCA = Montreal Cognitive Assessment; NIHSS = National Institute of Health Stroke Scale; NMES = Neuromuscular Electrical stimulation; NR = Not reported; OD = Oropharyngeal dysphagia; PAS = Penetration-Aspiration Scale; PD = Parkinson’s disease; P-DHI = Persian Dysphagia Handicap Index; PDQ-39: Parkinson’s Disease Questionnaire-39; PNF = proprioceptive neuromuscular facilitation; RCT = Randomised Controlled Trial; SLP: Speech-Language Pathology; SRS = Swallowing Rating Scale; SSA = Standardized Swallowing Assessment; SIS-6 = Swallowing Impairment Score; SWAL-QOL = Swallow Quality-of-Life Questionnaire; tDCS = transcranial Direct Current Stimulation; UC = Usual Care; VDS = Video-fluoroscopic Dysphagia Scale; VFSS = Video-Fluoroscopic Swallowing Study; WHO = World Health Organisation; WST = Water Swallow Test; TWST = Timed Water-Swallow Test.

**Table 3 jcm-11-00685-t003:** Outcome of behavioural interventions for people with oropharyngeal dysphagia.

Study	Intervention Goal	Intervention Agent, Delivery and Dosage ^a^	Materials and Procedures ^a^	Outcome Measures	Treatment Outcome ^a^
Ayres et al. [19]	To verify the effectiveness of a manoeuvre application in swallowing therapy in patients with PD.	*Intervention agent:* NR *Dosage:* Experimental group: chin-down manoeuvre and swallowing orientation: 4 sessions per week (30 min each). Orientation group: Swallowing orientation only: 4 sessions per week (30 min each).	Three groups: Experimental group: Chin-down posture manoeuvre (patient instructed to ‘swallow lowering the head until chin touches in the neck’). Patients performed manoeuvre twice a day, swallowing saliva, during meals, throughout the week, at home. Patients were given a form to record the number of times the manoeuvre was performed at home. Patients also given instructions for optimal feeding and swallowing related to ‘swallowing orientations’: (1) environment during feeding (2) posture (3) meal-time (4) oral hygiene. Written instructions given. Orientation group: Patients also given instructions for optimal feeding and swallowing related to ‘swallowing orientations’: (1) environment during feeding (2) posture (3) meal-time (4) oral hygiene. Written instructions given. Control group: No intervention received during 4-week period. Written instructions given.	*Primary outcomes:* FEES; Clinical evaluation (checking 21 signs and symptoms of oropharyngeal dysphagia and rating these as present or absent); FOIS; SWAL-QOL.	Experimental group showed significant improvement in clinical evaluation of dysphagia compared to two other groups regarding solid (*p* = < 0.001) and liquid (*p* = 0.022). Analysis of FEES did not show differences between groups. Experimental group presented with significant improvement in scores of domains frequency of symptoms (*p* = 0.029) and mental health (*p* = 0.004) on the SWAL-QOL when compared with the groups that did not receive intervention.
Carnaby et al. [20]	Compare standard low-intensity and high-intensity behavioural interventions with usual care (UC) for dysphagia	*Intervention agent:* Speech pathologist (Low/high intensity); physician and speech pathologist when referred (UC) *Dosage (average):* Swallowing sessions = 8.1, treatment days = 15.3, duration of session = 21.6 min	*UC (control):* Physician management. Patient referred to hospital speech pathology if needed. Treatment- feeding supervision, safe swallowing. If prescribed–VFSS. *Standard low-intensity:* Swallowing techniques, environmental modifications (upright for feeding); safe swallowing advice (eating rate); dietary modification (speech pathologist, 3 times per wk for 1 month. Strategies VFSS. *Standard high-intensity:* Direct swallowing exercises (effortful swallow, supraglottic swallow technique), dietary modification (from speech pathologist, daily for 1 month. Swallowing exercises established by examination and VFSS.	*Primary outcomes:* return to pre stroke diet < 6 months *Secondary outcomes:* time to return to normal diet, proportion recovered, functional swallowing, dysphagia-related complications, died, were institutionalised, or dependent in daily living 6 months post stroke.	Compared with usual care and low-intensity therapy, high-intensity therapy was associated with an increased proportion of patients who returned to a normal diet (*p* = 0.04) and recovered swallowing (*p* = 0.02) by 6 months.
Carnaby et al. [21]	Effectiveness and safety of exercise based swallowing therapy and neuromuscular electrical stimulation for dysphagia	*Intervention agent:* NMES & MDTP-Speech pathologists, >5 years dysphagia experience. UC-experienced therapist *Dosage:* 1 h/day × 3 wks (15 sessions)	*McNeill Dysphagia Therapy Program (MDTP):* Exercise-based swallowing-criteria for initial oral bolus materials for therapy and advancement on 11-step “food hierarchy”. Simple swallowing. Clinicians monitor each swallow. *Neuromuscular Electrical stimulation (NMES):* VitalStim^®^-Active NMES/sham, common single electrode placement-midline above hyoid bone to superior to cricoid cartilage)-ascending amplitude until amplitude reached. *Usual care treatment control (UC):* Behavioural swallowing treatment strategies common in dysphagia treatment.	Primary outcomes: Ability to swallow (MASA), oral intake (FOIS). Secondary outcomes: Barium swallow outcomes, self-perceived swallowing, weight, time to pre-stroke diet, complications.	Post treatment dysphagia severity significant between groups (*p* ≤ 0.01). MDTP greater change vs. NMES or UC for increased oral intake (*p* ≤ 0.02), functional outcomes at 3-mnths (RR = 1.7, 1.0–2.8), earlier time for “return to pre-stroke diet” (*p* < 03).
Choi et al. [22]	Effects of Shaker exercise on aspiration and oral diet	*Intervention agent:* Caregiver (SE), occupational therapist (CDT) *Dosage:* 30 min/day, 5 days/wk × 4 wks	*Shaker Exercise (SE):* Isometric and isokinetic movements. 3 head lifts held for 60 s in supine; 60 s rest. 30 reps head lifts observe toes without raising shoulders-without hold. *Conventional Dysphagia Therapy (CDT):* Orofacial muscle exercises, thermal tactile stimulation, therapeutic/compensatory manoeuvres.	Primary outcomes: PAS from VFSS. *Oral diet level* by FOIS.	Experimental group greater improvement on PAS (*p* < 0.05) and FOIS (*p* < 0.05) vs. control group.
DePippo et al. [23]	Effect of graded intervention on occurrence of dysphagia related complications	*Intervention agent:* Dysphagia therapist (SLP?) *Dosage:* Bi-weekly session monitoring for all groups	*Group A*–Patient-managed diet. One session-therapist recommended diet based on MBS results and compensatory swallowing techniques. Patient chose diet (regular vs. graded). *Group B*–Therapist-prescribed diet (MBS) and swallowing techniques, evaluated every other week. *Group C*-Therapist prescribed diet and daily reinforcement of swallowing techniques through mealtime dysphagia group.	Primary outcomes: *Dysphagia related complications:* Pneumonia, dehydration, calorie-nitrogen deficit, recurrent upper airway obstruction, and death.	No significance between groups for time until end inpatient stay or to 1-year post. Only significance was patients in group B developed pneumonia sooner than group A.
Eom et al. [24]	Effect of resistance Expiratory Muscle Strength Training (EMST) on swallowing function	*Intervention agent:* NR *Dosage:* 5 days p/wk × 4 wks, 5 sets of 5 breaths on device × 25 p/day. Both groups treatment 30 min × 5 days/wk × 4 wk	*Experimental group (EMST + Conventional treatment):* Portal Expiratory Muscle Strength Trainer (EMST150). Patients opened mouth after inhalation, EMST mouthpiece between lips. Blew strongly and rapidly until pressure release valve within EMST device opens. Pressure release set to open if pressure target exceeded. < 1-min break after each session, for muscle fatigue and dizziness. *Placebo group (Sham EMST + Conventional treatment):* Trained using a sham non-functional EMST device with no loading device. Conventional treatment.	Primary outcomes: VDS and PAS based on a VFSS to analyse oropharyngeal swallowing function.	Experimental significant in VDS pharyngeal phase (*p* = 0.02 and 0.01) and PAS vs. placebo (*p* = 0.01). Both significant VDS all phases (all *p* < 0.05). Experimental only significant in PAS (*p* = 0.01 vs. 0.102).
Gao and Zhang [25]	Effects of rehabilitation training on dysphagia and psychological state	*Intervention agent:* NR *Dosage:* 3 sessions/actions performed morning, midday and evening. 7 days p/wk × 42 days	All patients received routine treatment including internal medicine, traditional rehabilitation and routine nursing. *Control:* Traditional tongue and mouth exercises. Each movement repeated 10 times as one session. *Shaker exercise:* Supine position, single action raised head to look at feet. 30 reps = set of actions. Perform 3 sets of actions-continuously or with 1-min relaxation until complete. (Denoted as ‘Goa & Zhang, 2017a’ in Figure 5.) *Chin Tuck Against Resistance (CTAR) exercise:* Patients seated tucking chin to compress inflatable rubber ball for 30 reps = set of actions. Perform 3 sets, continuously or with relaxation. (Denoted as ‘Goa & Zhang, 2017b’ in Figure 5.)	Primary outcomes: *Dysphagia:* VFSS at baseline, 2, 4, 6 wks post. Swallowing function, PAS *Psychological state:* Self-Rating Depression Scale (SDS) baseline, 6 wks post.	Degrees of dysphagia improvement, between 2–4 wks in CTAR and Shaker. Significantly higher in CTAR (87%) and Shaker (77%) vs. control (43%) (all *p* < 0.05). Significantly lower SDS in CTAR vs. Shaker/control 6 wks post (all *p* < 0.05).
Guillén-Solà et al. [26]	Effectiveness of inspiratory/expiratory muscle training (IEMT) and neuromuscular electrical stimulation (NMES)	*Intervention agent:* Occupational, speech, physical therapist *Dosage:* Control- 3 hrs p/day × 5 days wk × 3 wks. Group *2-*2 × p/day, 5 days × 3 wks. Group *3–*40-min daily sessions (5 days per wk × 3 wks)	*Control/SST:* Multidisciplinary inpatient rehabilitation for mobility, activities of daily living, swallowing and communication. Education self-management of dysphagia, oral exercises and compensatory techniques based on VFSS. *EMST + SST:* Inspiratory/Expiratory Muscle Training (EMST)-respiratory training, 5 sets of 10 respirations, 1 min unloaded recovery breathing, with therapist. Pressure 30% of maximal expiratory pressures increased weekly. *NMES + Sham EMST + SST:* Sham respiratory muscle training, fixed at 10 cmH2O. Neuromuscular electrical stimulation using VitalStim device. Supervision by speech therapist, electrodes on suprahyoid muscles 80 Hz of transcutaneous electrical stimulus, patients to swallow when felt muscle contraction.	Primary outcomes: *Dysphagia severity* by PAS. *Respiratory muscle* *strength* (maximal inspiratory and expiratory pressures). Post- and 3-month follow-up.	Maximal respiratory pressures most improved Group 2: treatment effect 12.9 (CI 4.5–21.2) and 19.3 (CI 8.5–30.3) for maximal inspiratory and expiratory pressures. Swallowing security improved in Groups 2 and 3. PAS and complications -no between group difference 3-months.
Hägglund et al. [27]	Effect of oral neuromuscular training among older people in intermediate care with impaired swallowing	*Intervention agent:* Dental hygienists and speech pathologist *Dosage:* NR	*Intervention (IQoro*^®^ + Usual care)*:* The device IQoro^®^ was used for oral neuromuscular training. The device is designed to stimulate sensory input and strengthen the facial, oral, and pharyngeal muscles. Professionals provided training instructions. If participants had difficulties performing training, staff or family members were instructed on how to assist. *Control (Usual care):* Usual care with adjustments in food consistencies and posture instructions.	Primary outcomes: Swallowing rate (timed water swallow test) Secondary outcomes: Signs of aspiration during water swallow, swallowing related quality of life (QOL).	Swallowing rate significant improvement, intervention vs. controls post (*p* = 0.01), 6 months following (*p* = 0.03). Aspiration significantly reduced in intervention vs. controls (*p* = 0.01). QoL no between-group differences
Hägglund et al. [28]	To determine the effects of neuromuscular training on swallowing function in patients with stroke and dysphagia.	*Intervention agent:* Discipline NR *Dosage:* Neuromuscular training = 3 times per session and 3 times daily before eating Orofacial sensory vibration stimulation was performed 3 times daily before meals. 5 weeks of training in total.	Group A-Orofacial sensory-vibration stimulation: Patients received 5 weeks of continued oro-facial sensory vibration stimulation using an Oral B^®^ electric toothbrush. Instructions given on how to stimulate the buccinator mechanism, lips, external floor, tongue. Group B-Orofacial sensory-vibration stimulation + oral neuromuscular training (Muppy^®^): Patients received oral neuromuscular training for 5 weeks + oro-facial sensory vibration stimulation 1) Oral device (Muppy^®^) was used for oral neuromuscular training that aims to stimulate sensory input and strengthen facial, oral, pharyngeal muscles. Muppy^®^ is placed pre-dentally behind closed lips and pt sits in upright position. Patients hold device against a gradually increasing horizontal pulling force for 5–10 s whilst trying to resist the force by tightening the lips (2) oro-facial sensory stimulation of buccinator using electric toothbrush. Verbal, practical and written instructions about training given. Patient/caregiver reported training in a log-book. All patients in both groups self-administered or were assisted by relatives or ward staff in oro-facial sensory vibratory stim.	*Primary outcome:* Changes in swallowing rate measured by the Timed Water Swallow Test (TWST). *Secondary outcomes:* changes in lip force measured by lip-force test + swallowing dysfunction as measured by VFS (in lateral projection).	Swallowing rate: After intervention, both groups had improved significantly (Group B, *p*< 0.001; Group A, *p* = 0.0001) in TWST, but no significant between-group difference in swallowing rate. At 12 month follow-up, Group2 had improved significantly in swallowing rate compared to Group A (*p* = < 0.032) Lip force: Significant improvement in lip force in Group 2 (*p* < 0.001) compared to non-significant improvement in Group 1 (*p* = 0.079). Improvement in Group 2 maintained at 12 month follow up.
Hwang et al. [29]	Effect of tongue stretching exercises (TSE) on tongue motility and oromotor function in patients with dysphagia after stroke.	*Intervention agent:* TDT/TSE by occupational therapists. *Dosage: TSE–*5 × p/wk × 4 wks. Stretching 20 × p/ day.	*Control group*: Traditional Dysphagia Treatment (TDT)- oral facial massage, thermal-tactile stimulation, compensatory skill straining. Both groups received TDT. *Experimental group*: +Tongue Stretching Exercise (TSE); dynamic/static stretching exercises (20 reps each). Dynamic-therapist pulled patient’s tongue to end feel point of ROM and held for 2–3 s before guiding back to mouth. Static-therapist pulled tongue to end feel point, held 20 s.	Primary outcomes: *Oromotor function*-Oral phase events of VDS, VFSS *Tongue motility*-Distance from lower lip to tip of tongue during maximum protrusion of the tongue.	Experimental significant differences in tongue motility, bolus formation, tongue to palate, bolus loss, oral transit time-oral VDS phase (*p* < 0.05 for all). Control significant for lip closure only (*p* < 0.05).
Jakobsen et al. [30]	Effect of the intensification of the nonverbal facilitation of swallowing on dysphagia.	*Intervention agent:* Occupational therapist *Dosage:* 30 sessions (10-min rest, 20-min session, 10-min rest), 3 wks (2 × day).	*Experimental treatment:* Facial Oral Tract Therapy (F.O.T.T.) concept-rehabilitation intervention using structured tactile input and nonverbal facilitation techniques (to allow for effective function in meaningful daily life activities). *Control group:* Treatment comprised stimulating activities in the facial oral tract similar to those of the intervention group but without facilitation of swallowing or verbal request to swallow.	Primary outcomes: FOIS, PAS, and electrophysiological swallowing specific parameters (EMBI).	Intervention feasible. PAS and FOIS improved in both groups, no group differences. Swallowing specific parameters reflected clinically observed changes.
Jang et al. [31]	Effects of Mechanical Inspiration and Expiration (MIE) exercise using mechanical cough assist on velopharyngeal incompetence	*Intervention agent:* NR *Dosage:* 20 sessions Both groups, 30 min 2 × day, 5 × wk × 2 wks.	*Study group MIE exercise:* CNS-100 Cough Assist^®^ and conventional swallowing rehabilitation. *Inspiration-* positive pressure 15–20 cm H2O, increased to 40 cm H2O for 2 s. *Expiration–*similar pressure 10–20 cm H2O above positive pressure; held 3–6 s, simulating airflow during cough. Patient coordinated respiratory rhythm to cough assist machine. *Control:* Conventional dysphagia rehabilitation of oral motor and sensory stimulation, NMES, oral exercises for safe swallow.	Primary outcomes: *Swallowing function* American Speech-Language-Hearing association scale, functional dysphagia score, and PAS, VFSS. *Coughing function*-peak cough flow.	Study group significant improvement in functional dysphagia score- nasal penetration degree. Nasal penetration degree and peak cough flow showed greater improvement in study vs. control group.
Jeon et al. [32]	To investigate the effects of NMES plus upper spine cervical mobilisation on forward head posture, and swallowing in stroke patients with dysphagia.	*Intervention agent:* Joint mobilisation was performed by a physical therapist (with over 160 h of manual therapy education. NMES was delivered by 3 experienced OTs. *Dosage:* once a day, 3 × times a week, for 4 weeks; both groups received NMES for 30 min; experimental group received 10 min of upper cervical spine mobilisation; control group received 10 min of sham mobilisation.	All interventions were performed in sitting position. *NMES:* *Intervention group* received upper cervical spine (C1–2) mobilisation with NMES. *Mobilisation:* Therapist used one hand to hold the subject’s C1(atlas); other hand placed on subject’s occiput. Mobilisation force could not be standardised. NMES was applied to the suprahyoid using VitalStim^®^. Electrodes attached to the motor point of the suprahyoid muscles (digastric) to induce anterior excursion and vertical elevation movements of hyoid bone during normal swallowing. Stimulation was applied by gradually increasing the intensity to the level that patients felt a grabbing sensation in the neck without pain or laryngospasm. *Control group*: Patients received upper cervical spine sham mobilisation combined with NMES.	*Primary outcome:* Forward head posture measured by CCFT (Stabilizer ^TM^ Pressure Biofeedback) and craniovertebral angle (CVA). Swallowing function measured by VFS and PAS.	The intervention group showed significantly better scores in CCFT (*p* = 0.05) and in CVA (*p* = 0.05) than in control group. PAS scores were significantly better in the intervention group compared to control group (*p* = <0.05). Significant increase in VFS total score and PAS than in the control group (*p* = <0.05)
Kim et al. [33]	The effects of Proprioceptive Neuromuscular Facilitation (PNF) on swallowing function of stroke pts with dysphagia	Intervention agent: NR Dosage: PNF-based short neck exercises 3 times a week for 30 min each time for 6 weeks	*Experimental group: PNF* Patients started by lying on a bed with head and neck positioned off the bed (tester supported left laryngeal region with his right hand and placed left fingertips below patient’s jaw) Patient instructed to look at target object in a direction 15 degrees diagonally to the right side Tester then initiated given exercises by moving the patient’s neck in a diagonal direction opposite to the direction specified Patient instructed to ‘draw your jaw inward’ and tester applied a level of resistance to the patients jaw to fully activate neck flexor below jaw (rotation to the right) Same exercises applied in opposite direction. *Control group: Shaker exercise* 1. Isometric exercises: Patients lay on bed and raised their heads without moving shoulders off the bed, looked at ends of their feet for 60 s, and then lowered heads back on the bed. If patient had difficulty raising his/her head, they were asked to perform same exercise for 3 times for as long as they could. Isotonic exercises: Patients raised their head in same posture and looked at the ends of their feet 30 consecutive times.	*Primary outcome:* New VFSS and ASHA NOMS Scales.	Statistically significant improvements in: premature bolus loss, residue in the valleculae, laryngeal evaluation, epiglottic closure, residue in pyriform sinuses, coating of pharyngeal wall after swallowing, pharyngeal transit time and aspiration on both new VFSS scale and ASHA NOMS scale (*p* < 0.05). Control group also demonstrated statistically significant improvements in premature bolus loss, residue in the valleculae, laryngeal evaluation, epiglottic closure, residue in pyriform sinuses, pharyngeal transit time and aspiration (*p* < 0.05). No statistically significant differences between the groups were found in new VFSS scale and ASHA NOMS scale.
Kim and Park [34]	Effect of modified chin tuck against resistance (mCTAR) exercise on patients with post-stroke dysphagia.	*Intervention agent:* Occupational therapist *Dosage:* 30 min × 5 days a week, for 6 weeks	*Experimental group mCTAR exercise*: PhagiaFLEX-HF device. Subject seated, fixed part of device to desk, firmly attach chin surface under chin. Exercise performed in isotonic/isometric. Isometric- holding chin down for 10 s against resistance (10 s, 3 times). Isotonic-30 × reps chin-down against resistance. *Traditional dysphagia treatment (TDT):* Oral facial massage, thermal-tactile stimulation and compensatory training.	Primary outcomes: *Aspiration and oral diet* -PAS and FOIS. Secondary outcomes: Rate of *nasogastric tube remova*l was analysed.	Experimental statistically significant improvement in PAS and FOIS vs. control (*p* < 0.001). Rates of nasogastric tube removal were 25% (experimental) vs. 15% (control).
Koyama et al. [35]	Feasibility and effectiveness newly developed Modified Jaw Opening Exercise (MJOE) in poststroke patients with pharyngeal residue.	*Intervention agent:* Speech pathologist/physician *Dosage:* 4 × sets daily, 5 × p/wk × 6 wks. (6 s × 5 reps = 1 set)	*Intervention MJOE:* Surface electrodes mandibular midline. Participants closed mouth, sitting position, pressed tongue against hard palate. Trainer hand under participant’s chin and applied upward vertical resistance. Visual feedback given. Maintained 80% Maximum Voluntary Contraction (MVC). *Control sham exercise isometric jaw closing exercise:* Surface electrodes to masseter, visual feedback, 20% MVC.	Primary outcomes: VFSS was performed before and after exercise. The distance between the mental spine and the hyoid bone (DMH) and hyoid displacement (HD) were measured.	No temporomandibular joint or neck pain. Intervention group, DMH decrease where anterior HD ended and an increase in anterior HD were seen. Control, no changes.
Krajczy et al. [36]	Effects of dysphagia therapy in patients in the early post-stroke period.	*Intervention agent:* Physiotherapist *Dosage:* Physiotherapy program average 60 min × day, × 15 days	*Control/both groups:* Safe food education and neurological physiotherapy depending on patient dysfunction. Therapy included passive, assisted, supported and respiration exercises, erect posture, walking re-education, and training on NDT Bobath and PNF methods. *Study group:* +original dysphagia treatment, restoring chewing and swallowing functionality–Strengthening and breathing exercises and thermal stimulation.	*Primary outcomes:* *Swallowing function -* Timed test of swallowing *Swallowing reflux –* Controlled swallowing after swallowing blended food. Reflex categorised as good or delayed.	*Swallowing reflux, Cough and voice quality and swallowing time, number of swallows and* SpO2 All Statistically significant differences between groups after therapy (*p* = <0.01).
Kyodo et al. [37]	To evaluate the effectiveness of puree diets containing a gelling agent for the prevention of aspiration pneumonia in elderly patients with moderate to severe dysphagia.	*Intervention agent:* Gastroenterologists experienced in transnasal endoscopy along with a speech therapist evaluated swallowing. Discipline who created gelling agent (intervention) NR. *Dosage (average):* NR	Patients underwent endoscopic swallowing evaluation while sitting in a chair/sitting up in bed. Images of oropharynx and larynx were displayed on a monitor and recorded on digital video recorder. *Pureed diet without gelling agent* was made by mixing 100 g of white rice and 50 mL of water with a blender for one minute. Texture characteristics (IDDSI Level 4) were: hardness, 1760 ± 125 N/m^2^; cohesiveness, 0.59 ± 0.03; adhesiveness, 224 ± 56 J/m^3^. *Pureed diet with gelling agent* was made by mixing 100 g of rice porridge at > 70 degrees with 0.5 g of the gelling agent with a blender for one minute. Texture characteristics (IDDSI Level 4) were: hardness, 312 ± 11.3 N/m^2^; cohesiveness, 0.81 ± 0.02; adhesiveness, 108 ± 5.8 J/m^3^.	*Primary outcome:* Presence of material in throat using endoscopic cyclic ingestion score (0 to 4) *Secondary outcomes:* Sense of material remaining in the throat after swallowing of pureed rice and/or test jelly; degree of dysphagia using Hyodo-Komagane score (0 to 12: mild 0–3; moderate 4–7; severe 8–9)	Residuals in throat were significantly less likely with pureed rice with than without the gelling agent (median cyclic ingestion score (range); 1 (0–4) vs. 2 (0–4); *p* = 0.001. Irrespective of presence or absence of the gelling agent, the sense of materials in the throat was significantly less frequent in older patients (*p* = <0.01). No adverse events occurred.
Logemann et al. [38]	3 treatments for aspiration on thin liquids—chin-down posture, nectar-thickened liquids, or honey-thickened Liquids.	*Intervention agent:* Speech pathologist *Dosage:* NR	*Chin-down intervention:* chin to the front of the neck, three swallows of 3 mL of thin liquid from a spoon and three swallows of the same liquid from an 8-oz cup filled with 6 oz of liquid. *Nectar or* *Honey-thickened liquids:* on the two thickened liquid interventions, three swallows of 3 mL of thickened liquid from a spoon and three self-regulated swallows, performed as separate swallows, each from an 8-oz cup filled with 6 oz of the thickened liquid.	*Primary outcomes:* *Swallowing function-*VFSS	49% aspirated all interventions, 25% not any. More on thin liquids despite chin-down posturing vs. using nectar-(*p* < 0.01) or honey-thickened (*p* < 0.01). More on nectar- vs. honey thickened (*p* < 0.01).
Manor et al. [39]	Effectiveness of visual information while treating swallowing disturbances in patients with PD.	*Intervention agent:* Speech and swallowing therapist *Dosage:* Each group 5 × 30 min sessions, during 2-wk period and a 6th session 4 wks after the 5th one	*Control–conventional therapy*: Both interventions swallowing exercises and compensatory therapy based on FEES. Compensatory strategies carried out with different food and liquid consistencies in clinic, patient practiced at home. *VAST:* video-assisted tool during each session, for educating and assisting understanding structure of swallowing. Patients observed a normal swallowing process and their distorted one. After learning compensatory technique, patient practiced it during drinking and eating in the clinic after observing video then at home. During next four sessions patients observed video with suitable compensatory swallowing technique while eating and drinking focusing on the new swallowing behaviour.	*Primary outcomes:* *Swallowing function*-by fiberoptic endoscopic evaluation of swallowing (FEES). *Quality of life-*quality of care and degree of pleasure from eating assessed by questioners	Significant improvement in swallowing functions both groups. FEES significantly greater reduction in food residues in pharynx in VAST vs. conventional treatment group. SWAL-QOL scores significant between groups favour of VAST: burden, eating desire, social functioning, mental health, symptom frequency (*p* < 0.01).
Mepani et al. [40]	Effect of the Shaker exercise on thyrohyoid muscle Shortening improve pharyngeal dysphagia	*Intervention agent:* Speech pathologist *Dosage:* Biweekly 45-min therapy sessions for 6 weeks.	*Traditional therapy:* 5 times daily. Laryngeal and tongue ROM exercises and swallowing manoeuvres (Super-Supraglottic Swallow, Mendelsohn Manoeuvre, Effortful Swallow). Shaker Exercise: 3 times per day for 6 weeks. Isometric and isokinetic head-lift in supine position. Patients raised head high and forward to observe toes. Isometric–3 times head lifts held 60 s, 60-s rest period. Isokinetic-30 head lifts at constant velocity, performed without holding or rest periods.	*Primary outcomes:* *Change in thyrohyoid* *muscle shortening* by Videofluoroscopy	After therapy, the percent change in thyrohyoid distance in the Shaker Exercise group was significantly greater vs. traditional therapy (*p* = 0.034).
Moon et al. [41]	Effects of Tongue pressure strength and accuracy training (TPSAT) on tongue pressure strength, swallowing function, and quality of life in stroke patients with dysphagia.	*Intervention agent:* Occupational therapist *Dosage:* TPSAT and traditional dysphagia therapy 30 min × day; Only traditional therapy performed 30 min × twice daily. Both groups, daily 5× times wk × 8 wks.	Both groups received standardized physical/occupational therapies. *Traditional dysphagia therapy*: thermal tactile stimulation, Mendelsohn manoeuvre, effortful swallow, diet modification. *TPSAT with traditional dysphagia treatment:* TPSAT consisted of an anterior and posterior isometric tongue strength exercise and an isometric tongue accuracy exercise. The protocol involved five sets of tongue-to-palate presses, 6 reps per set for each session. Isometric tongue accuracy exercise, amplitudes were set at 50, 75, 100% of maximum pressure from first isometric strength. Participants generated precise pressures within 10 kPa error for each amplitude.	*Primary outcomes:* *Tongue pressure strength -*maximum isometric tongue pressures (MIPs) of anterior, posterior tongue using Iowa Oral Performance Instrument. *Swallowing function*-MASA; *QoL-*SWAL-QOL	TPSAT with traditional dysphagia significantly improved MASA, SWAL-QOL, and MIPs. Traditional dysphagia significantly increased MASA, SWAL-QOL, and MIPs anteriorly (*p* < 0.05). TPSAT significant in anterior, posterior MIPs, tongue movement MASA, vs. controls (*p* < 0.05).
Park et al. [42]	Effects of EMST on the activity of suprahyoid muscles, aspiration and dietary stages in stroke patients with dysphagia.	*Intervention agent:* Occupational therapist *Dosage:* 5 days × wk × 4 wks. 5 sets × 5 breaths on device, 25 breaths per day.	*Experimental group:* resistance set at 70% range of MEP (Maximal Expiratory Pressure). Subjects open mouth following maximum inhalation, EMST mouthpiece between lips, close mouth. Blow strong and fast until pressure release valve in EMST device opens- expiratory pressure exceeded set target. *Placebo group:* training using sham device-non-functional device, little effect of physiologic load on targeted muscles.	*Primary outcomes:* *Activity in the suprahyoid muscle group -*using surface electromyography (sEMG). PAS used to assess VFSS results. *Dietary stages-*FOIS.	Experimental significantly more in suprahyoid muscle activity (*p* = 0.01), liquid PAS (*p* = 0.03) and FOIS (*p* = 0.06), but not semisolid type PAS (*p* = 0.32), vs. placebo.
Park et al. [43]	Effect of chin tuck against resistance exercise (CTAR) on the swallowing function in patients with dysphagia following subacute stroke.	*Intervention agent:* Occupational therapist *Dosage:* 30 min × day, × 5/wk, × 4 wks	*CTAR:* Isometric CTAR, patients chin tuck against device 3 × 60 s no repetition. Isotonic CTAR, patient 30 reps by strongly pressing against resistance of the device and releasing it. Therapist demonstrated exercise methods. *Conventional dysphagia treatment:* Both groups -orofacial muscle exercises, thermal tactile stimulation, and therapeutic or compensatory manoeuvres.	*Primary outcomes:* *Swallowing function* -Functional Dysphagia Scale (FDS) and PAS, based on VFSS	Experimental more improvement in oral cavity, laryngeal elevation/epiglottic closure, residue in valleculae, and residue in pyriform sinuses of FDS and PAS compared vs. controls (*p* < 0.05, all).
Park et al. [44]	Effects of Effortful Swallowing Training (EST) on tongue strength and swallowing function in patients with stroke.	*Intervention agent:* Occupational therapist *Dosage:* Training 30 min, 5× days per wk × 4 wks. Both groups conventional dysphagia treatment 30 min/day, 5 days/wk × 4 wks.	*Experimental EST:* Patients pushed tongue onto palate, squeezing neck muscles, swallow forcefully. Performed 10 times p/session, 3 sessions p/day. Effortful swallowing confirmed by therapist through visual observation and palpation. *Control group:* Swallow naturally without intentional force. Patients given small spray of water to induce swallowing, and rest. Both groups received conventional dysphagia therapy (compensatory techniques -chin tuck, head tilting, rotation; therapeutic techniques -orofacial muscle exercises, thermal tactile stimulation using ice sticks, expiratory training).	*Primary outcomes:* *Tongue strength-*Iowa Oral Performance Instrument. *Oropharyngeal swallowing function* VDS, based on VFSS.	Experimental group greater improvements in anterior and posterior tongue strength vs. control (*p* = 0.05 and 0.04), and greater improvement in oral phases of VDS (*p* = 0.02).
Park et al. [45]	Effects of game-based Chin Tuck against resistance exercise (gbCTAR) and head-lift exercise on swallowing function and compliance in dysphagia post-stroke	*Intervention agent:* Occupational therapist *Dosage:* 5 × wk × 4 weeks. Traditional dysphagia treatment (TDT) 30 min per day	*Experimental group:* performed gbCTAR exercise LES 100 device. Before gbCTAR exercise, 1-RM measured for resistance values. 1-RM, resistance bar placed directly beneath jaw, and chin tuck directed against resistance. gbCTAR exercise at threshold of 70% 1-RM, divided into isometric and isotonic exercises, combined with the game. *Control group:* head lift exercises in supine (isometric and isotonic). Both groups TDT- oral facial massage, thermal-tactile stimulation and compensatory training.	*Primary outcomes:* *Swallowing function-*VDS and PAS. *Dietary assessment-*FOIS *Compliance with the 2 exercises*-(motivation, interest, physical effort, fatigue), numerical rating self-report scale.	No significant between group difference in VDS, PAS, FOIS. Compliance, motivation and interest Scores significantly higher, and scores for physical effort needed and fatigue significantly lower, in experimental vs. control.
Park et al. [46]	Effect of Resistive Jaw Opening Exercise (RJOE) on hyoid bone movement, aspiration, and oral intake level in stroke patients.	*Intervention agent:* Occupational therapist *Dosage:* 30 min × 5 times wk × 4 wks.	*Experimental group:* RJOE device to provide resistance to suprahyoid muscles. Isometric exercise, 30 s with device resistors pressed downward (3 times, 30–60 s of rest). Isotonic exercise repeatedly depressed by RJOE by holding device resistance down for 2–3 s then returned to original state (10 reps, 3 sets) with 30 s rest. *Placebo group:* RJOE using 1-mm thick device with almost no resistance to suprahyoid muscles. Exercise type and frequency of RJOE same as experimental group. Both groups received conventional dysphagia therapy after intervention, which involved orofacial muscle exercises, thermal tactile stimulation and therapeutic or compensatory manoeuvres.	*Primary outcomes:* *Hyoid bone movement* -by two-dimensional analysis of anterior and superior motion on VFSS. *Aspiration-*PAS *Oral intake level*-FOIS.	Both groups significant differences in hyoid movement, PAS, FOIS (*p*< 0.05). No significant difference between groups except for liquid type, PAS. Effect sizes (Cohen’s d) 0.6–1.1 for anterior, superior movement of hyoid bone, semisolid and liquid type of PAS, and FOIS respectively.
Ploumis et al. [47]	Evaluate cervical isometric exercises in dysphagic patients with cervical spine alignment disorders due to hemiparesis after stroke.	*Intervention agent:* Allied health *Dosage:* inpatient 12 wks, speech 30 min daily. Experimental-4× reps 10 min, 3× day, 12 wks.	All patients -inpatient program including physiotherapy, occupational and speech therapy. Speech included deglutition muscle strengthening, compensatory techniques. *Experimental group:* +plus cervical isometric strengthening exercises contract neck muscles under resistance forward-backward-sidewards). *Control group:* Regular speech therapy plus sitting balance.	*Primary outcomes:* Cervical spine radiographs in erect (sitting/standing) position coronal, sagittal C2-C7 Cobb angle, VFSS to evaluate deglutition.	Experimental group- more pronounced correction (*p* < 0.01) of cervical alignment in both planes and greater improvement (*p* < 0.05) of deglutition too, than control group.
Sayaca et al. [48]	Whether combined isotonic technique of Proprioceptive Neuromuscular Facilitation (PNF) is superior to Shaker exercises in improving function of swallowing muscles.	*Intervention agent:* Shaker ‘CS’ (?). PNF physiotherapist *Dosage:* Each exercise set 1 x per day, 3x wk x 6 wks.	*Shaker exercises:* isometric (3 reps) and isotonic contractions (30 reps) neck flexor muscles. Patients raised head to observe toes without raising shoulders. Isometric- lifted head, held for 1-min 3 times, 1-min rest. Isotonic- lifted head 30 reps, no holding. *PNF:* Combined isotonic technique- concentric, stabilizing and eccentric contraction without relaxation. Stabilizing contractions to improve control, force, coordination, and eccentric contraction. Moved head against resistance with open mouth- kept position for 6 s against resistance in seated position; kept position while physiotherapist moved back to initial position. 30 reps per day.	*Primary outcomes:* *Swallowing difficulties* Turkish Eating Assessment Tool (T-EAT-10). *Capacity, volume, and speed of swallowing-*100 mL-water swallow test. *Contraction amplitude changes*-motor unit activity, by superficial electromyography.	T-EAT-10 decreased both groups (*p* < 0.001). Water swallowing capacity and volume improved both groups (*p* < 0.001). No change in swallowing speed both groups (*p* > 0.05). Maximal voluntary contraction of suprahyoid muscles higher in PNF vs. Shaker (*p* < 0.05).
Steele et al. [49]	Compare outcomes of two tongue resistance training protocols	*Intervention agent:* Speech pathologist *Dosage:* 24 sessions (TPPT or TPSAT), 2–3× wk, 8–12 wks. 60 tongue-pressure tasks per session.	*Tongue-pressure profile training (TPPT):* emphasized pressure-timing patterns that are typically seen in healthy swallows by focusing on gradual pressure release and saliva swallowing tasks. *Tongue- pressure strength and accuracy training (TPSAT):* emphasized strength and accuracy in tongue-palate pressure generation and did not include swallowing tasks.	*Primary outcomes:* *Posterior tongue strength, oral bolus control, penetration– aspiration and vallecular residue-* VFS, PAS	Both groups significant tongue strength and post-swallow vallecular residue with thin liquids. Stage transition duration (bolus control), PAS no significant differences.
Tang et al. [50]	Effect of rehabilitation therapy on radiation-induced dysphagia and trismus in nasopharyngeal carcinoma (NPC) patients after radiotherapy.	*Intervention agent:* Therapists, assistants *Dosage:* Rehabilitation group, exercises 3× per day, each 15 cycles, 45 cycles per day.	Both groups routine treatment. *Rehabilitation group:* training by therapists at hospital, continued at home post-discharge by exercise booklet, guardian oversight and calendar Exercises: *Tongue-*range of motion exercises included passive and active movement exercises. *Pharynx and Larynx-*exercises changing body position to maximize swallow function and minimize aspiration. Swallow manoeuvres included effortful swallow and Mendelsohn manoeuvre. Sensory procedures utilizing pharyngeal cold stimulation performed by therapists. *Exercise for Trismus-* Active jaw movements- opening/closing mouth repeatedly, opening mouth slightly, moving lower mandible to left and right, stretched chin downward and forward and a range of passive jaw movements. *Control group:* No rehabilitation exercises Both groups received routine treatment (e.g., anti inflammatory treatment for aspiration pneumonia)	*Primary outcomes:* *Severity of dysphagia*- water swallow test *Trismus-*LENT/SOMA score and the interincisor distance (IID).	Rehabilitation group only significant improvement in swallowing function. Percentage of patients with effective results in rehabilitation higher than control (*p* = 0.02). Control IID significantly decreased at Post (*p* = 0.001), both groups decreased at 3 months, rehabilitation group less than controls (*p* = 0.004). Trismus in rehabilitation higher vs. control (*p* = 0.02).
Tarameshlu et al. [51]	Effects of Traditional Dysphagia Therapy (TDT) on swallowing function in Multiple Sclerosis (MS) patients with dysphagia.	*Intervention agent:* Therapist *Dosage:* both groups 6 weeks, 18 sessions, 3 × per week, every other day.	*Traditional Dysphagia Therapy (TDT):* Includes oral motor control, range of motion exercises, swallowing manoeuvres, strategies to heighten sensory input. *Usual care (UC):* postural changes, modifying volume and speed of food presentation, changing food consistency and viscosity, and improving sensory oral awareness.	*Primary outcomes:* *Swallowing ability-* Mann Assessment of Swallowing Ability (MASA) *Secondary outcomes:* PAS and PRRS.	Groups improved MASA, PAS and PRRS (*p* < 0.001). All significantly greater in TDT vs. UC group. Large effect size MASA in TDT (d = 3.9) and UC (d = 1.1).
Troche et al. [52]	Treatment outcome of device-driven EMST on swallow safety, physiologic mechanisms through measures of swallow timing and hyoid displacement.	*Intervention agent:* Clinician *Dosage:* EMST, 4 weeks, 5 days per week, for 20 min per day, using a calibrated or sham, handheld device.	*Expiratory muscle strength training (EMST):* device set to 75% of participant’s average MEP. Visited weekly by clinician-instructed to wear nose clips, deep breath, hold cheeks lightly, blow hard into device, identify air was flowing freely through device (once reached threshold pressure). *Sham:* Sham device identical to EMST device, pressure release valve non-functional and to 75% of participants’ average MEP-no physiologic load to muscles.	*Primary outcomes: S**wallow function*-judgments of swallow safety, PAS scores, swallow timing, and hyoid movement from VFS images.	EMST improved swallow safety, PA scores vs. sham. EMST improvement of hyolaryngeal function during swallowing, findings not evident for sham group.
Wakabayashi et al. [53]	Effects of resistance training of swallowing muscles in community dwelling older individuals with dysphagia.	*Intervention agent:* Research co-workers *Dosage: intervention* exercises for 10 s; 1 set = 10 reps. 2 sets per day 3× per wk × 3 months	*Control/both groups:* dysphagia brochure (about oral hygiene, tongue resistance exercise, head flexion exercise against manual resistance, nutrition, and food modifications). *Intervention:* resistance exercises for swallowing muscles involving tongue resistance exercise and head flexion against manual resistance. Research co-workers instructed participants once how to perform resistance training.	*Primary outcomes:* *Improvement in dysphagia* -Eating Assessment Tool (EAT-10) score. *Secondary outcomes:* Tongue pressure	Percentage of participants with EAT-10 scores <3 not statistically significantly different between groups *p* = 0.6). Post intervention EAT-10 (*p* = 0.7) and mean tongue pressure (*p* = 0.4).
Woisard et al. [54]	Effect of a personalised transportable folding device for seating on dysphagia	*Intervention agent:* Occupational therapy *Dosage:* 1 x training session with device (D+ group) and without device (D- group).	*D-/All groups:* All patients training session: evaluation of needs, impact of head positioning on swallowing, adapted position of head through body positioning, practice using occupational therapy cushions or personalised transportable folding device for seating (DATP) according to randomisation. *D+ group:* in charge to determine characteristics of the device required so they could have them during the training session. Instruction for patients was to put the personalised instructions into practice by using the device.	*Primary outcomes:**quality of swallowing* *Secondary outcomes:**posture, device acceptability, QoL*. Measurement of hyoid bone movement during swallowing. VFSE and questionnaire.	Significantly better posture both groups (*p* < 0.001), more hyoid bone motion in D+ group. Significant mean difference for D+ group vs. D− group, for horizontal and vertical movement. Other swallowing markers not significant.
Zhang and Ju [55]	Clinical improvement of nursing intervention in swallowing dysfunction of elderly stroke patients.	*Intervention agent:* Nursing staff *Dosage:* NR	*Control group:* conventional nursing service that strictly conforms to the doctor’s advice. *Nursing intervention:* (1) Psychological intervention, nurses communication with patients/family, evaluates psychological state, encourages and comforts. (2) Health education, nurse introduces knowledge about swallowing dysfunction and effects through videos and images. (3) Rehabilitation exercises, pronunciation training, muscle training, mouth opening exercises, ingestion training. (4) Diet intervention, appropriate foods should be chosen according to specific conditions.	*Primary outcomes: S**wallowing dysfunction–*30 mL water drink test *Living quality-*assessment questionnaire of living quality (GQOL-74), includes physical, psychological and social functions, and material life. *Pulmonary infection–*rate *Nursing satisfaction*–self-made questionnaire.	Improved swallowing dysfunction higher in intervention vs. control (*p* < 0.05). Scores of physical, psychological and social functions, and material life and nursing satisfaction higher in intervention vs. control (*p* < 0.05). Pulmonary infection lower in intervention vs. control *p* < 0.05).

^a^ Terminology as by authors. *Notes*. CVA = cerebrovascular accident; EMST = Expiratory Muscle Strength Training; FEES = Fiberoptic Endoscopic Evaluation of Swallowing; FOIS = Functional Oral Intake Scale; MASA = Mann Assessment of Swallowing Ability; MBS = Modified Barium Swallow; MIE= Minimally Invasive Oesophagectomy; MDTP = McNeill Dysphagia Therapy Program; MEP = Maximum Expiratory Pressure; NMES = Neuromuscular Electrical stimulation; NR = Not reported; OD = Oropharyngeal dysphagia; PAS = Penetration-Aspiration Scale; PD = Parkinson’s disease; P-DHI = Persian Dysphagia Handicap Index; PNF= Proprioceptive Neuromuscular Facilitation; PRRS = Pharyngeal Residue Rating Scale; QoL = Quality of life; RCT = Randomised Controlled Trial; SIS-6 = Swallowing Impairment Score; SWAL-QOL= Swallow Quality-of-Life Questionnaire; VDS= Video-fluoroscopic dysphagia scale; VFSS = Video-Fluoroscopic Swallowing Study; TWST= Timed Water-Swallow Test; VDS = Videofluoroscopic Dysphagia Scale; VFSE = Videofluoroscopic Examination.

**Table 4 jcm-11-00685-t004:** Between subgroup meta-analyses comparing intervention groups of included studies.

Subgroup	Hedge’s *g*	Lower Limit CI	Upper Limit CI	*Z*-Value	*p*-Value
*Intervention type*					
Combined vs. CDT (Combined) (*n* = 5)	0.610	0.263	0.957	3.446	0.001 *
Combined vs. CDT (Compensation) (*n* = 3)	1.180	0.362	1.998	2.828	0.005 *
Rehabilitation vs. CDT (Combined) (*n* = 1)	0.019	−0.656	0.659	0.057	0.955
Rehabilitation vs. CDT (Rehabilitation) (*n* = 3)	0.178	0.304	1.133	3.395	0.001 *
Rehabilitation vs. No CDT (*n* = 3)	0.842	0.440	1.244	4.110	<0.001 *
*Selected interventions*					
Shaker vs. CDT (*n* = 2)	1.038	0.300	1.776	2.756	0.006 *
CTAR vs. CDT (*n* = 3)	1.045	0.427	1.663	3.316	0.001 *
EMST vs. no CDT (*n* = 2)	0.819	0.389	1.250	3.733	<0.001 *
*Diagnostic groups*					
Acquired Brain Injury (*n* = 1)	0.947	−0.247	2.141	1.554	0.120
Parkinson’s disease (*n* = 1)	0.792	0.273	1.311	2.898	0.003 *
Stroke (*n* = 13)	0.731	0.474	0.988	5.573	<0.001 *
*Outcome measures*					
Superior hyoid displacement (*n* = 1)	0.994	−0.124	2.112	1.743	0.081
MASA (*n* = 2)	0.512	−0.574	1.599	0.925	0.355
PAS (*n* = 11)	0.804	0.572	1.036	6.789	<0.001 *
Tongue motility oromotor function (*n* = 1)	0.359	−0.470	1.189	0.849	0.396

*Notes*. * Significant.

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
