# Peer review of "Behavioural Interventions in People with Oropharyngeal Dysphagia: A Systematic Review and Meta-Analysis of Randomised Clinical Trials"

_jcm, 2022, doi:10.3390/jcm11030685_

Round 1

Reviewer 1 Report

Congratulations to all authors, this very relevant systematic review contains a comprehensive list of evidence based behavioral interventions in oropharyngeal dysphagia. This review is important as it shows the gaps in research, but also highlights the evidenced based treatment methods available in clinical practice. Methodology is described consciously, although sometimes, the overall picture through the big tables (2 and 3) becomes less clear.

Most references are recent (within the last 5 years), based on publication dates it might be possible to moderate the tables and reduce the content in tables 2 and 3.

Figure 4 and 5 contain a lot of information, hard to read in the actual lay-out. I’m not sure these entire figures are necessary in your main article.

Some specific questions and remarks:

  • P3 line 106: why do you exclude dysphagia following radiotherapy?
  • Table 2/3: maybe redistribution from the different studies based on pathology/population, or behavioral treatment is more appropriate? I believe that study characteristics and outcome can’t be reported separately in order to correctly interpret the results. Changing between both tables is rather uneasy while reading the article.
  • P8: the study from Hashemian is executed in a population post-thyroidectomy, but I understood that patients post thyroidectomie were excluded? (p3 line 106)
  • P10: the study from Mepani is executed in a mixed-population stroke and CRT, I suggest to exclude this study to take into account your exclusion criteria (p3 line 106)
  • P11: a typo was found in the description of the diagnosis in the study by Ploumis (Hemip7aresis)
  • P11: the study from Tang is executed in population post radiotherapy, which I think is in contradiction with your inclusion criteria
  • P21 line 521, 253: didn’t you exclude this patients (post-thyroidectomy and post-radiotherapy)?
  • P24 line 325: what do you mean with ‘two studies were excluded to reduce heterogeneity between studies? One study focusses on mobilization and the other one on resistance training. Why do they introduce heterogeneity?
  • Discussion:
    • A lot of studies are focusing on stroke-population. Did you take post-onset time into account during your analyses?
    • What is your opinion about treatment time and frequency? Conclusion/statements about both aspects of behavioural treatment are important for clinicians and patients considering treatment planning and reimbursement.

Reviewer 2 Report

This manuscript presents a systematic review and meta-analysis on the efficacy of behavioural interventions for oropharyngeal dysphagia. Overall, it is a well-written and comprehensive review with a nice flow and detailed descriptions of methodology and presentation of findings. Despite the ambiguity in terminology and definition of behavioural dysphagia interventions across studies in the literature, the authors managed to categorise and analyse their effects in a coherent and organised manner. My comments are detailed below:

  1. The use of the term, “conventional dysphagia therapy (CDT)”, is vague and diverse across studies. How were the data analysed if CDT is used as a control for one study but as an experimental condition for another? For example, in Gao and Zhang (2016)’s study, tongue and mouth exercises were used as “conventional intervention” (control), but in Moon et al (2018)’s study, tongue strengthening exercise, although not the same exercise regime, was used as experimental condition.
  2. Page 4, line 158: Were the sample sizes corrected for studies where there is more than one comparison with the same group (i.e., Treatment A vs Treatment B vs Control)?   
  3. Page 4, line 181: Why were these interventions selected?
  4. Table 2: Moon et al. (2018) and Ploumis et al. (2018): Please specify what was used as control.
  5. Table 2, footnote: Typo: “US” should be “UC”
  6. Table 3: Choi et al. (2017) and Kim & Park (2019): Please add CDT to the experimental groups as well.
  7. Table 3: Krajczy et al. (2019): Is “original dysphagia treatment” the same as CDT? If so, please change it to CDT for better consistency.
  8. Page 22, line 291: Typo: “or motor” should be “oromotor”.
  9. Figure 3: What does yellow circle with exclamation mark represent?
  10. Table 4 and Figure 5: According to the descriptions in Table 3, the CDT used in the studies by Choi et al. (2017), Hwang et al. (2019) and Kim & Park (2019) are not compensatory as some included oromotor exercises and thermal-tactile stimulation. Please clarify if this is accurate.

Round 2

Reviewer 2 Report

I would like to thank the authors for addressing my previous concerns. I have one minor suggestion: It may be helpful to add the reasons for selecting those specific interventions for further analysis (response 3), i.e. based on commonalities across studies, to the Meta-analysis subsection. 
